## Special feature reviews

evolution, theoretical biology

microbiome, symbiosis, vertical and horizontal transmission, mixed modes of transmission, mutualism

**Authors for correspondence:**
Philip T. Leftwich
e-mail: p.leftwich@uea.ac.uk
Tracey Chapman
e-mail: tracey.chapman@uea.ac.uk

Special Feature paper: Application of ecological and evolutionary theory to microbiome community dynamics across systems. Guest edited by Dr James McDonald, Dr Britt Koskella, Professor Julian Marchesi.

## PUBLISHING

# Transmission efficiency drives host–microbe associations

Philip T. Leftwich[1], Matthew P. Edgington[2] and Tracey Chapman[1]

[1]School of Biological Sciences, University of East Anglia, Norwich Research Park, Norwich, Norfolk NR4 7TJ, UK
[2]The Pirbright Institute, Pirbright, Woking, Surrey GU24 0NF, UK

PTL, 0000-0001-9500-6592; MPE, 0000-0003-1922-9529; TC, 0000-0002-2401-8120

Sequencing technologies have fuelled a rapid rise in descriptions of microbial communities associated with hosts, but what is often harder to ascertain is the evolutionary significance of these symbioses. Here, we review the role of vertical (VT), horizontal (HT), environmental acquisition and mixed modes of transmission (MMT), in the establishment of animal host–microbe associations. We then model four properties of gut microbiota proposed as key to promoting animal host–microbe relationships: modes of transmission, host reproductive mode, host mate choice and host fitness. We found that: (i) MMT led to the highest frequencies of host–microbe associations, and that some environmental acquisition or HT of microbes was required for persistent associations to form unless VT was perfect; (ii) host reproductive mode (sexual versus asexual) and host mate choice (for microbe carriers versus non-carriers) had little impact on the establishment of host–microbe associations; (iii) host mate choice did not itself lead to reproductive isolation, but could reinforce it; and (iv) changes in host fitness due to host–microbe associations had a minimal impact upon the formation of co-associations. When we introduced a second population, into which host–microbe carriers could disperse but in which environmental acquisition did not occur, highly efficient VT was required for host–microbe co-associations to persist. Our study reveals that transmission mode is of key importance in establishing host–microbe associations.

## 1. Introduction

Rapid advances in sequencing technologies and bioinformatic analyses are revolutionizing the understanding of microbial communities that live on, in or near plant and animal hosts. Microbial symbionts can play crucial roles in many aspects of an organism's biology [1] and associations between hosts and microbes can vary from commensal and parasitic relationships, through to obligate mutualisms in which the fate of microbe and host are intimately entwined [2]. Microbes can be intracellular, extracellular, co-inherited with their hosts, horizontally transmitted or acquired from the environment. Crucial to the outcome of host–microbe associations is the mode of microbial inheritance and the mechanisms that allow recurrent co-association of microbes and their hosts [3].

In this article, we focus on animal (mostly insect) microbe systems, review modes of transmission of microbes to their hosts and discuss their likely evolutionary significance (table 1). We also explore the as yet under-researched selective potential of mixed modes of transmission (MMT). To assess the relative importance of the different factors contributing to host–microbe associations, we develop mathematical models at the population level, between a single host and microbe species, and evaluate:

(i) which modes of transmission favour long-term associations between microbes and animal hosts;
(ii) whether host reproductive mode and host mate choice can promote host–microbe associations;

**Table 1.** Summary of key terms used within this study and a description of their implementation within the mathematical models. Full details on the mathematical implementation of each mechanism are given in the electronic supplementary material.

| mechanism | biological definition | model representation |
|---|---|---|
| vertical transmission (VT) | the passage of the microbe from the host mother (sometimes the father) to its offspring | a percentage of offspring from carrier mothers inherit the microbe |
| environmental acquisition | asocial acquisition of independently proliferating microbes available in the environment, habitat and/or host diet | a probability that non-carrying individuals acquire the microbe following birth but prior to sexual maturity. Independent of population-level carrier density; this is a steady-state characteristic within the models |
| horizontal transmission (HT) | the non-vertical passage of microbes among hosts, may include larval conspecific feeding or sexual transmission | a probability of carriers transmitting to non-carriers, calculated as a multiple of the total carrier frequency; therefore, HT is dependent on population-level carrier density |
| mixed modes of transmission (MMT) | any combination of transmission modes including VT, HT and environmental acquisition by which microbes can be acquired by hosts | any stable carrier frequency which occurs because of more than one (exclusive) mechanism of microbial transmission |
| fitness | measured as individual reproductive success, and equal to the contribution to the gene pool of the next generation made by individuals of the specified genotype/phenotype | the probability of an individual surviving to sexual maturity relative to the non-carrier level (0.5 means half the number of individuals survive, whereas 2 indicates twice as many individuals survive) |
| dispersal | when individuals move from one site to another to mature or breed | a percentage (2%) of each population is exchanged at random each generation; this occurs following uptake/acquisition and fitness but prior to mating |
| host mate choice | the process that occurs whenever the effects of traits expressed in one sex lead to non-random matings with the opposite sex; either (i) assortative mating, i.e. preference for similar characteristics (microbial carrier/non-carrier), or (ii) preference for a consistent phenotype, i.e. for microbial carriers | $n$-choice framework—the choosing sex samples $n$ potential mates; if a preferred type is found, then mating occurs or else mating is with the $n$-th sampled individual |

(iii) if host–microbe associations require a host fitness benefit to establish at a high population frequency; and

(iv) whether parameters leading to a high frequency of host–microbe associations alter when considering one homogeneous population versus two partially isolated populations.

## (a) Modes of co-transmission of hosts and their microbes

Host-associated microbial communities are ubiquitous, and the bacteria within them can show transient or resident associations. There are three general and non-exclusive mechanisms by which microbes can be introduced to an animal host: (i) the passage of a symbiont from the host mother to offspring (vertical transmission or VT); (ii) environmental acquisition, usually through ingestion of microbes with diet, though this may also occur via mechanism (iii) horizontal transmission (HT), i.e. by consumption of microbes shed into the environment by conspecifics and/or by social/sexual transmission. These mechanisms have been reviewed in detail elsewhere [4,5], we discuss these briefly, below,

along with the as yet under-researched evolutionary potential for mixed modes of transmission (MMT).

### (i) Vertical transmission of symbiotic microbes

Vertically transmitted symbionts necessarily exhibit strong host fidelity, arising from their direct transfer from mother to offspring. Many insects carry heritable microbes [2,6,7] including the widespread intracellular symbionts *Rickettsia*, *Cardinium* and *Wolbachia*. These microbes infect the host germline and become incorporated within maturing oocytes [6,7]. The high fidelity of VT has facilitated the transition of many heritable symbionts to obligate associations, leading to the loss of their ability to propagate independently [2]. Such microbes are reliant solely on maternal VT to spread, with their fitness being dependent on the survival and propagation of their female hosts [8].

A classic example is found within the aphids and their maternally inherited obligate endosymbiont *Buchnera aphidicola* [9]. Here, the bacteria provide the host with essential metabolites and are housed within a specialized structure in the host gut. However, VT does not always indicate mutualism between host and microbe, as evidenced by the many and varied effects of the parasitic intracellular symbiont

*Wolbachia. Wolbachia* strains manifest a wide variety of reproductive manipulations of hosts, such as feminization, parthenogenesis and most commonly, cytoplasmic incompatibility (CI) [7]. CI results in sperm-egg incompatibility between infected males and uninfected females and drives *Wolbachia* through host populations, as uninfected females have a reproductive disadvantage relative to those that are infected [7].

Maternal inheritance is not restricted to intracellular symbionts, and there are increasing reports of symbiotic gut bacteria being transmitted from mother to offspring [10,11]. If widespread, this could open up new opportunities for evolutionary co-associations to form. As gut microbes are extracellular, they are unlikely to be transmitted through oocytes (though one such example has been reported in the wax moth *Galleria mellonella* [12]). Despite this, several direct and indirect routes for VT have been identified, including by (i) contact smearing of microbes onto the egg surface during or after oviposition [13], (ii) oviposition site inoculation and reingestion by offspring [14], (iii) coprophagy [15] and (iv) social acquisition from parent to offspring [16].

### (ii) Environmental acquisition or horizontal transmission of symbiotic microbes

For many insects, the majority of their symbionts are residents of the gut microbiome, which is composed mostly of bacteria acquired via contact, in or on, ingested food. However, the microbiome also has the potential to be structured in various ways, e.g. by host diet selection [17,18], screening [5,19–21], gut physiology [22] or by spatial compartmentalization of microbial species within the gut [10]. Sequencing of 16 s rRNA genes has shown that insect hosts typically house relatively simple gut communities (often dominated by a few key taxa [18]) with the majority of the gut microbiome composed of non-specialist microbes [23,24]. These tend to be newly acquired from the external environment each generation [19,25–27].

Differences between the species composition of gut microbiomes and the communities of the host's external environments suggest that at least some form of microbiome structuring occurs [17,28]. The ability to describe gut microbiomes and hosts across space and time (phylosymbiosis) has led to much interest in the roles and functional significance for hosts of their microbiomes [29,30]. There are two key questions to consider when evaluating the evolutionary significance of host–gut microbiome relationships [31]. The first is the extent to which host–microbiome associations are assembled randomly or deterministically [5]. The second is whether there is the potential for recurrent environmental acquisition of specific symbionts at sufficient fidelity for them to have the potential to shape the evolution of their hosts [3]. In general, we expect environmental acquisition or HT to be less efficient than VT, lowering the likelihood that host–microbe associations will evolve [5,10,32,33]. However, specific conditions may nevertheless allow environmental acquisition or HT to favour selection for host–microbe co-associations.

HT may occur when juvenile hosts acquire microbes through a source pool of parental or conspecific microbes. If there are fitness benefits gained by hosts, then microbe carriers will increase each generation as the ratio of carriers to non-carriers, and therefore the number of contributors of microbes has increased. Alternatively, mechanisms that manipulate hosts or promote transmission fidelity might also promote associations. For example, one route by which the transmission fidelity of HT bacteria might increase is via mate choice for individual hosts with a similar microbiome. In *Drosophila melanogaster,* assortative mating by diet, and thus microbiome similarity, is reported [34]. Different gut microbiota is associated with differences in cuticular hydrocarbon profiles [35] that may influence the expression of mating pheromones [36]. The finding of positive assortative mating by microbiomes, however, shows a lack of replicability [37–40]. Nevertheless, control of host behaviour by symbionts, particularly when this influences transmission dynamics, has the potential to influence the strength of co-associations [41–45]. It has been proposed that influences on host mate choice effects by transient gut microbiota, such as a preference for carriers of a particular microbe to only mate with other carriers of the same microbe, can act as a precursor to reproductive isolation and thus speciation [46].

In theory, within newly isolated populations, key bacteria could be routinely 'added back to the pool' for acquisition by juvenile hosts (VT or HT), while the introduction of new microbes is restricted by host mating exclusion. However, to our knowledge, no experimental or theoretical models have yet addressed how bacteria-induced host mate choice can act as a driver of RI within a homogeneous population or reinforce isolation between populations. This is an omission that we tackle here by developing new theory, as described below.

Alternatively, host–microbe relationships with asocially environmentally acquired bacteria might occur if hosts can screen in or out specific bacterial species or functional traits [5,19–21]. Environmentally acquired bacteria could then play a role in shaping host evolution, even in the absence of VT or HT. Consistent with this, an increasing number of studies show that, when controlling for diet and environment, it is often possible to align the microbiome with that of its host's evolutionary history [29,30]. This suggests that the non-stochastic assembly of microbiomes occurs through environmental acquisition. The strength of selection for such host–microbe co-associations is likely to be strongly affected by whether microbial selection is primarily host-led [47,48] or due to microbial niche selection [49].

Host-led selection, in particular, is thought to promote parallelism/phylosymbiosis at a functional, but not necessarily taxonomic, level [50,51]. For example, host physiology may be adapted to the metabolic functions of resident microbiomes, thus directing the functional properties of microbes that can colonize them. The mechanisms involved may include the actions of host antimicrobial peptides [52], immune genes [53] and host-specific biofilms that affect gut colonization efficiency [54]. Host-led control of colonization could also be influenced by the morphological and physio-chemical conditions in the host gut [10]. Phylosymbiosis would then result because the guts of closely related host species provide similar environmental niches and thus favour the formation of microbiomes that echo evolutionary phylogeny, independent of any host fitness benefits or close evolutionary co-associations.

Studies reporting the existence of phylosymbiosis in host–gut microbiome associations are increasing. However, in general, microbial detection methods, in isolation, cannot separate residents from dead or transient microbes [55], potentially leading to overestimates of phylosymbiosis. Bacteria may

also sometimes be repeatedly acquired from the environment into non-primary hosts, which would represent a microbial evolutionary dead-end. There are well-verified examples in which a resident microbiome seems mostly lacking [56]. For example, in an investigation of inter- and intra-specific variability in microbial biomass in caterpillars, Hammer *et al.* [56] observed an extremely low density of recurrent bacteria in guts in comparison to those found on the hosts' food. Given current detection methods used and the near ubiquity of laboratory contaminants, transients, parasitic and pathogenic microbes, it seems likely that there are more cases of hosts lacking resident or beneficial microbiomes than has so far been realized.

### (iii) Mixed modes of transmission of symbiotic microbes

In highly specialized intracellular host–microbe associations, such as the *Buchnera*, pea aphid system, host and microbe phylogenies match with high fidelity [57] and the genomes of the obligate symbionts display evolutionary signatures of VT, namely a reduction in genome size and a loss of metabolic capabilities [9]. However, it is thought that even for obligate symbiotic bacteria, propagation exclusively through VT is comparatively rare [11], and in many cases, there is instead evidence for current or previous mixed modes of microbial transmission. For example, the intracellular symbiont *Wolbachia* exhibits strong VT, yet has a considerably larger genome than would be predicted on the basis of evolved co-dependence with its hosts [58,59], and discordant phylogenies between *Wolbachia* species and their hosts suggests an extensive evolutionary history of HT [60,61].

By contrast, gut microbiomes, previously perceived as transient, may exhibit greater stability than expected. For example, the *D. melanogaster* fruit fly microbiome has long thought to be formed from communities of microbes living on recently ingested food [62,63]. However, it also appears that some bacterial strains may form stable associations in a host-specific manner [64,65]. For example, *Acetobacter thailandicus* appears to form a permanent association with the *D. melanogaster* gut once it becomes established through environmental acquisition. However, it can also propagate to offspring (VT) and conspecifics (HT) through continuous bacterial shedding [65].

At least some symbiotic microbes, therefore, may be subject to mixed modes of transmission (MMT), involving combinations of VT, HT and environmental acquisition [4,11,14,65–67]. Bacteria that are capable of independent replication and proliferation in the environment but are also capable of colonization of the insect gut and VT, gain multiple opportunities for spread within hosts. A combination of transmission modes may thus greatly increase the range of ecological conditions that support symbionts [11,68]. For the host, there is also growing evidence that having some degree of flexibility around a core microbiome may aid in rapid diet-switching and localized adaptation [69].

Specific mechanisms of transmission are not necessarily required for MMT. For example, bacterial films surround insect eggs from contact smearing during oviposition and bacteria here may independently replicate to establish free-living independently replicating populations and/or be ingested by neonate larvae (and thus represent VT) and their conspecifics (HT) [14]. The lack of specialism and relaxed ecological constraints required may make this a frequent mechanism for symbiotic transmission [2,14,70]. MMT may also include transmission in a social context (e.g. parental care or social interactions such as mating) where the conditions of bacterial transfer through mechanisms such as egg-smearing may be predominantly vertical, horizontal or both depending on the social environment [4]. While MMT has the potential to reduce the strength and consistency of VT and selection for tight co-associations between microbes and hosts, it may have other benefits that increase the host fitness across a wide range of ecological conditions if hosts are able to supplement the microbiome with microbes from novel environments [66,69]. For these reasons, MMT is starting to garner increasing attention [4,11,70], though empirical and theoretical evidence for it is so far limited. We address this here by developing a theory that evaluates combinations of different symbiont transmission mechanisms.

## 2. Using models to test the effect of alternative transmission modes, host reproduction and microbe fitness benefits on the frequency of microbe-carrying hosts

While there are verbal models of induced host mate choice manipulation on host-associated microbes, to our knowledge, there are as yet no theoretical models that specifically explore the scenarios in which transient host–microbe associations can lead to established residencies with hosts, which then spread at a population level. Here, we address this omission by developing a series of hierarchical, deterministic, discrete generation mathematical models of the population-level frequency of host–microbe associations. In turn, we varied the mode of microbe acquisition (VT, HT or environmental acquisition), host reproductive mode (sex/asex), host mate choice (assortative mating or carrier preference) and magnitude of microbe effects on host fitness—
all over a range of realistic parameter space.
In doing so, we addressed four main questions:

(i) Which modes of transmission, either singly (VT, HT or environmental acquisition) or in combination (MMT), are most likely to lead to high frequencies of association between microbes and hosts?

(ii) Can modes of host reproduction (sexual or asexual) and host mate choice (positive or negative assortment for microbe carriers) increase the frequency of host–microbe associations?

(iii) Do associations between host and microbes require the microbial partner to increase host fitness in order to increase population frequencies of carriers?

(iv) Do parameters leading to a high frequency of host–microbe associations alter when considering one homogeneous population space when compared to two partially isolated populations?

These models allowed us, in a stepwise fashion, to predict whether stable host–microbe relationships would establish, first in a single homogeneous population, and then within a second, partially isolated population. We chose this population structure scenario because it allowed us to model explicitly a suggestion from the literature, that mate choice linked to microbiome carrier status can represent a strong precursor of reproductive isolation [38–40]. We restricted our approach to deterministic modelling due to the number of distinct modes

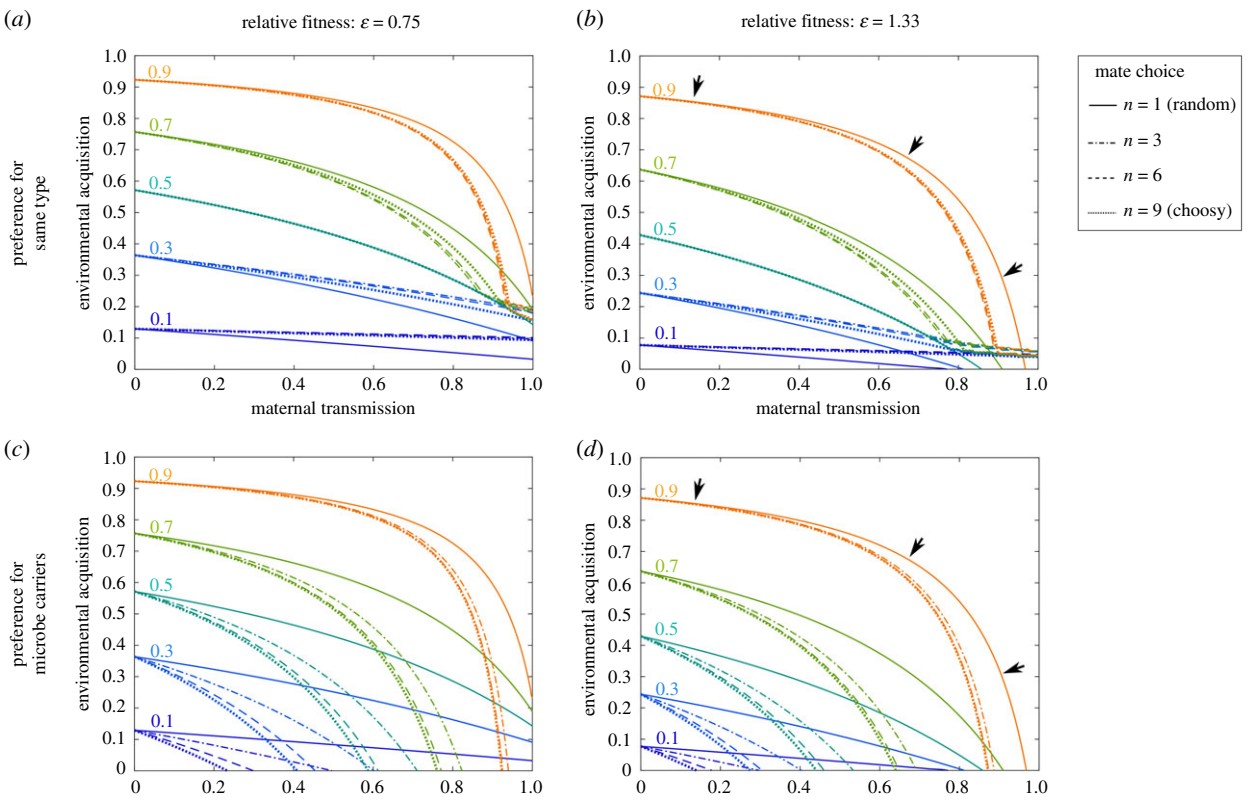

**Figure 1.** High-frequency transmission of microbes (through a variety of mechanisms—MMT) is a primary determinant of host–microbe carrier frequencies within a single population of sexually reproducing individuals. Each line represents different steady-state microbe carrier frequencies (0.1–0.9, increasing in 0.2 intervals). Multiples of each line represent additional degrees of male mate choice, e.g. the number of opportunities a male has to find a preferred mating partner from $n = 1$, (no choice) to 9 (choosiest). Black arrows in $(b,d)$ are used to highlight the main differences in conditions leading to a high carrier frequency (>0.9). Each panel represents a possible combination of host mate preference and relative fitness levels of host–microbe carriers ($\varepsilon = 0.75$ in $(a,c)$, versus $\varepsilon = 1.33$ in $(b,d)$) as specified. In all cases, anything less than perfect 100% maternal VT required some environmental acquisition of the microbe for carriers to reach a high frequency within the population. A comparison of $(a,b)$ shows that the relative fitness of host–microbe carriers compared to non-carriers had only a modest effect on the conditions required for carriers to reach high frequency. Positive assortative host mating by males for the same host type (carrier–carrier versus non-carrier–non-carrier) slightly changed the transmission conditions and either relaxed or increased transmission requirements based on overall carrier proportions. However, overall, the actual strength of host assortative mating had only a minimal effect. A comparison of $(c,d)$ shows that preference by males for female microbe carriers had a modest effect on relaxing the stringency of transmission for carriers at all population frequencies. Effects of the microbes on host fitness again had minimal effect on the conditions required for carriers to reach high frequency. For full model outputs, see electronic supplementary material, figures S5a and S6b. In each case, here and for figure 2, equilibrium carrier frequencies of zero are not possible for most parameter combinations due to the model structure considered (instead frequencies asymptote toward zero). Carrier frequencies equal to one are attained but only in cases with perfect vertical transmission and/or environmental acquisition meaning they overlie figure axes and are not visible. (Online version in colour.)

of microbe transmission we wished to consider. This allowed us to disentangle the impacts of each parameter more easily. We also reasoned that, in considering a sufficiently large population, stochastic effects would only play a significant role when microbe frequencies are very low. The models and results were developed and run in MATLAB (R2016a; The MathWorks Inc., Natick, MA). A summary of model parameters is in table 1, while full details of modelling methods, effects and parameters tested are described in the electronic supplementary material. The full set of results for all parameters tested are shown in electronic supplementary material, figures S2–S6, and we summarize the key results in the main text, below.

### (i) Single population model: effects of transmission mode, host reproductive mode, host mate choice and microbe effects on host–microbe carrier frequency

Under scenarios of reduced or neutral fitness, when maternal VT was anything less than 100%, some degree of environmental acquisition, or horizontal transmission was required to promote a high frequency of microbe-carrying

hosts (figure 1a; electronic supplementary material, figures S2–S6 show the influence of horizontal transmission effects of $\tau = 0, 0.25$ and $0.5$ for all scenarios modelled). This effect was less pronounced when host–microbe carriers had higher fitness than non-carriers, and in this scenario imperfect (but nevertheless high) maternal VT could still lead to a high (>90%) frequency of carriers within the population (figure 1b).

The overall effect of host reproductive mode (sexual versus asexual) on host–microbe carrier frequency was minimal (compare figure 1 with electronic supplementary material, figure S2a). Within models including sexually reproducing hosts, the type and strength of mate choice had modest effects on transmission dynamics. When males had a positive assortative preference for mating with the same type of host (carrier:carrier/non-carrier:non-carrier), it generally decreased the transmission efficiency needed for high microbe carrier frequencies to be reached. However, at the highest levels of assortative mating, these parameter ranges were reduced as it became harder for the microbe to pass from carriers to non-carriers (figure 1a,b; electronic supplementary material, figure S6a). Preference for carriers, in

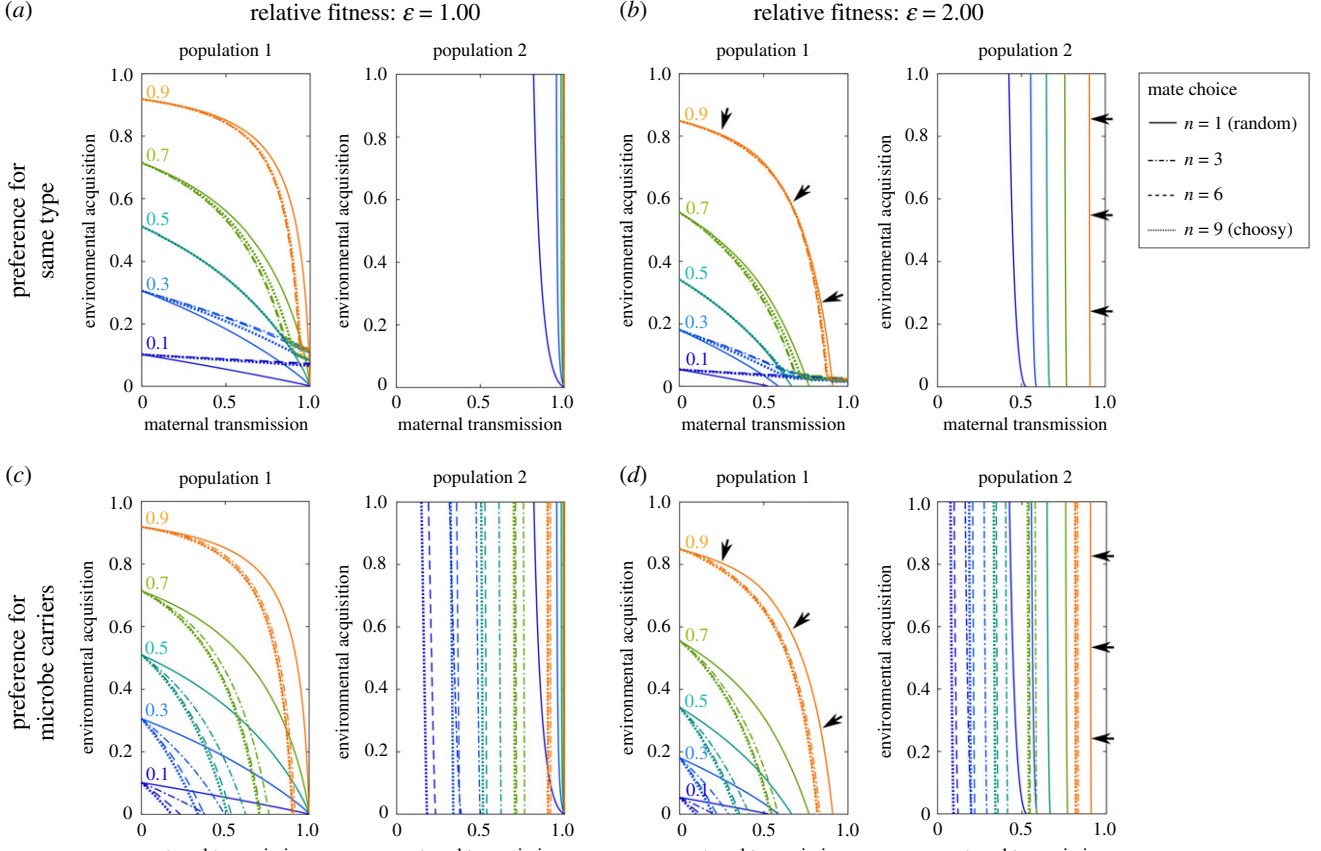

**Figure 2.** Host-associated microbes cannot establish in a new population in which they are incapable of independent proliferation outside hosts, unless VT is high, and they increase host fitness significantly. In population 1, microbes can proliferate independently of hosts and there is environmental acquisition by the host. In population 2, microbes cannot be acquired by hosts from the environment, only via maternal VT from dispersers from population 1. Each line represents different steady-state microbe carrier proportions (0.1–0.9, increasing in 0.2 intervals). Multiples of each line represent additional degrees of host mate choice, e.g. the number of opportunities a male has to find a preferred mating partner from $n = 1$ (no choice) to 9 (choosiest). Black arrows in (b,d) are used to highlight the main differences in parameter combinations capable of producing a high carrier frequency (>0.9). Each figure shows a possible combination of host mate choice and host–microbe carrier fitness relative to non-carrier fitness, with details as specified in the figure labels. Here, the microbe cannot spread with its host carriers into a second population at anything less than equal fitness to non-microbe-carrying hosts. Positive assortative mating by hosts for the same carrier type (carrier–carrier versus non-carrier–non-carrier) in (a,b) strongly reinforces population separation. In (c, d) preference by males is for female microbe carriers and this has the opposite effect, relaxing the degree of VT required for carriers to persist in population 2. For full model outputs, see electronic supplementary material, figures S5b and S6b. (Online version in colour.)

which all males preferred to mate with female carriers, had a modest effect on increasing the range of environmental acquisition and maternal transmission parameters that could result in high microbe carrier frequencies (>90%).

Female mate choice had minimal effect on carrier frequencies (compare electronic supplementary material, figures S4a,b with figures S3a,b). This is because females mediate VT, hence the choice of partner (carrier or non-carrier) does not alter transmission (compare electronic supplementary material, figures S3a–S5a and figures S4a–S6a). The overall effects of mate preferences were minimal in comparison to the requirements for high maternal VT.

Surprisingly, in these models, the effect of the microbe on host fitness had little impact upon the conditions required for carriers to reach high frequency (figure 1a versus 1b and figure 1c versus 1d; for the full range of fitness from 0 to 2 across all scenarios modelled, see electronic supplementary material, figures S2–S6).

Note that in all cases, there was substantial variation in the time taken to reach a microbe carrier frequency greater than 0.9 (from an initial carrier frequency of 0.01) ranging from one to approximately 250 generations depending on the precise parameter combinations.

**(ii) Two population model: effects of transmission mode, host reproductive mode, host mate choice and microbe effects on host–microbe carrier frequency**

When we included a neighbouring population that did not naturally host the microbe (with 2% bidirectional rate of dispersal), we found that the microbe could not spread into this second population under any set of transmission parameters, if the microbe reduced carrier fitness. For the microbe to spread, the fitness of carriers had to be at least equal to non-carriers. A high fitness advantage to microbe carriers somewhat compensated for imperfect maternal VT and a high level of maternal VT was still required for microbes to spread in the second population (electronic supplementary material, figure S2a versus S2b; figures S3b, S4b, S5b, S6b). The effect of host reproductive mode (sexual versus asexual) on host–microbe carrier frequency was again minimal (electronic supplementary material, figure S2b). In the sexual scenarios, mate preference for the same carrier host type reinforced the separation of the two populations (figure 2a). Even when microbe carriers were fitter than non-carriers, this type of assortative mating prevented the spread of the microbe into the second population (figure 2b). By contrast,

when all male hosts had a preference for mating with female microbe carriers, increasing mate choice promoted an increase in host–microbe carriers. The parameters of maternal VT required for the microbe to establish at a high frequency in the second population were relaxed (figure 2c,d). Increasing levels of HT allowed microbes to spread more easily into the second population (electronic supplementary material, figures S5b and S6b) under both mechanisms of mate choice.

This interesting result shows that host mate choice based on the presence of gut microbes can potentially reinforce or weaken reproductive isolation, depending on the type of host-mating preference expressed. The nature of this effect may depend on whether novel environmental microbe acquisition or HT predominates in specific animal systems.

## 3. Discussion and conclusion

Overall the modelling results revealed that the spread of microbes at a high frequency within a host population was more easily attained when there were high fidelity transmission routes and mixed modes of transmission that incorporated both maternal VT, HT and environmental acquisition. This supports the idea that the importance of such mixed transmission modes may have been overlooked [68].

Surprisingly, rather than host–microbe associations being strongly contingent upon benefits to host fitness and mutualism, the models suggest that transmission efficiency was the primary determinant of host–microbe carrier frequency. Furthermore, efficient microbial transmission, using one or several mechanisms, led to high host–microbe carrier frequencies even in the presence of a slight detriment to host fitness caused by microbe carrying.

A growing body of studies focuses on the existence of distinct modes of bacterial transmission and the effect of microbes on host physiology and fitness. However, what is less clear is the relative importance of these factors. In particular, widely held assumptions that the presence of a recurrent association in host–microbe interactions, or physiological effect on hosts, necessarily indicate the presence of a mutualism, need to be challenged. We explored here key factors in the establishment of host–microbial relationships by analysing population-level models of co-association.

The results suggested that neither phenotypic nor behavioural changes in the host (e.g. due to host mate choice for carriers) had a significant bearing on the transmission efficiencies required to promote a high frequency of host–microbe carriers. The overall effect of host reproductive mode (sexual/asexual) was minimal in our models. Within the sexual scenarios, mate choice for host–microbe carrier status, within a single population, had little effect on host carrier frequency and is thus unlikely by itself to lead to reproductive isolation.

Surprisingly, our results also suggested that the relative fitness of host carriers versus non-carriers was less important for increasing host–microbe carriers than the existence of efficient microbial transmission [1]. This is not to say that effects of microbes on the fitness of their hosts were absent. However, over a large parameter space of relative fitness from 0.75 to 2.0, the frequency of host–microbe carriers hardly changed. Though this finding underlies our general conclusion that microbe effects on hosts had a modest effect,

we note that such fitness effects gained in importance when we included dispersal, and hence there may be scenarios that we did not explore here, in which microbe–host effects have higher relative importance. If there is a strong fitness benefit to an association, it is possible that there could be selection on higher fidelity of transmission. Future research should also seek to verify these models, e.g. by using fluorescently labelled bacteria in combination with whole community microbial analyses, to track transmission and fitness benefits across generations and check that the reported outcomes are not unduly impacted by inaccurate model assumptions. At present, we lack suitable proofing data due to the absence of previous empirical or theoretical tests. However, our hope is that this initial theory will prove useful as a guide to frame additional experiments, as outlined above, to provide future verifications.

When we extended our models to consider a second population, both the effects of microbes on their hosts and maternal VT became more important. When microbes were incapable of independent replication (and hence environmental acquisition) in the second population, the introduction of microbes was possible only via migration of host carriers from the first population. Microbial spread was then only possible if the microbes significantly increased the fitness of host carriers relative to non-carriers and achieved high fidelity of transmission.

Effects of host-mating behaviour were also more pronounced in scenarios with a second population: microbes that induced a preference for same host-type matings reinforced pre-existing reproductive isolation caused by the low dispersal rate (2%), while an overall mating preference for carriers reduced reproductive isolation and slightly reduced the levels of VT required for spread. This suggests that direct effects of microbes on host mate choice cannot themselves result in reproductive isolation in otherwise homogeneous populations, but could reinforce or breakdown pre-existing isolation, depending on whether non-carriers prefer to avoid or seek out carriers (i.e. the form of host-mating preference), and with strong fidelity of microbial transmission. This shows that microbial transmission dynamics is key to models of microbe-induced RI and hence should not be overlooked [46], but see [31]. We suggest that tests of whether modest levels of VT, within partially isolated populations, where key bacteria are routinely 'added back' to their respective original populations, could be useful to establish long-term associations in which the introduction of competing microbes is restricted by host-mating exclusion. However, from our models, this effect appears to be moderated by the relative strengths of HT and environmental acquisition for microbe associations with juvenile hosts, and this could be tested empirically by combination approaches of microbiome community analyses and/or labelled bacterial strains to test their relative strengths in different model systems. Similar approaches could test the assumptions in our model that different transmission modes are routinely additive and explore whether, if multiple species/strains of microbes are considered, these might represent additional routes for microbe–microbe exclusion and competition.

Our approach here was based upon deterministic modelling, due to the number of distinct modes of microbe transmission we considered. This was an advantage in that it allowed us to easily disentangle the impacts of each parameter and we reasoned that stochastic effects would only play a significant role at low microbe frequencies. Given we were seeking

to explore mechanisms leading a microbe to spread to high frequency, we do not anticipate this approach to have had a significant or biasing impact on the results. However, we also recognize that these are preliminary models and that it would be useful in the future to specifically interrogate the effect of stochasticity. Another potentially useful addition, which lay outside of the initial modelling designs we deployed here, would be to include density-dependent effects. This would allow an exploration of the relative importance of an expanded range of ecological scenarios and to reveal interacting, nonlinear averaging, or synergistic effects as the frequency of microbe-carrying hosts increases.

Our results also contribute to tests of the hologenome concept [3], in particular to holobiont assemblage, transmission and mutualism [71]. An expectation of holobionts is that high partner fidelity leads to mutualisms from the establishment of collective fitness. However, here we found that host fitness may play a relatively minor role in the establishment of host–microbial interactions [71]. Our results are not incompatible with predictions concerning holobiont assemblage [67], as we show that there are multiple, potentially synergistic, mechanisms of host–microbe assembly. However, they do suggest that the role of functional effects on host performance may be less important than the ability to evade or downregulate host immune responses [30].

Data accessibility. The annotated MATLAB code for the modelling can be found at doi:10.5281/zenodo.3746199.

Authors' contributions. P.T.L, M.P.E and T.C. conceived the study; M.P.E. conducted the modelling; P.T.L, M.P.E and T.C wrote the manuscript.

Competing interests. We declare we have no competing interests.

Funding. This research was supported by funding from the BBSRC (BB/K000489/1, grant to T.C., Matthew Hutchings and P.T.L). M.P.E. is funded through a Wellcome Trust Investigator Award [110117/Z/15/Z].

Acknowledgements. We thank the editors of this special issue of *Proceedings B* for the opportunity to contribute to this volume, and the reviewers who commented on earlier drafts, and whose comments and suggestions helped greatly to improve and clarify this manuscript.

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
