## [Reviewer comments · Proceedings of the Royal Society B: Biological Sciences]

Review History

RSPB-2019-2052.R0 (Original submission)

Review form: Reviewer 1

Recommendation

Reject – article is scientifically unsound

Scientific importance: Is the manuscript an original and important contribution to its field?

Acceptable

General interest: Is the paper of sufficient general interest?

Acceptable

Quality of the paper: Is the overall quality of the paper suitable?

Acceptable

Is the length of the paper justified?

Yes

Should the paper be seen by a specialist statistical reviewer?

No

Do you have any concerns about statistical analyses in this paper? If so, please specify them explicitly in your report.

No

It is a condition of publication that authors make their supporting data, code and materials available - either as supplementary material or hosted in an external repository. Please rate, if applicable, the supporting data on the following criteria.

Is it accessible?

N/A

Is it clear?

N/A

Is it adequate?

N/A

Do you have any ethical concerns with this paper?

No

Comments to the Author

This paper contrasts different forms of transmission and explores their roles in driving host microbe associations to calibrate the role of environmental acquisition vs. vertical transmission in shaping observed associations at the scale of a single microbe / host. While I do think this effort (or something like it) is potentially of value, I don't think that the current model framework or manuscript achieves this. Perhaps more carefully addressing and expanding the range of contexts considered would be of greater value? For example one could build up a hierarchy of frames from asexual hosts, to sexual, to sexual with assortative mating, to sexual with disassortative mating, and spatial aggregation for example. This could also be done in a relatively methodologically consistent way via a series of nested matrix models (with the largest scale capturing two or more spatial patches, see some of Caswell's recent model extensions). The framing could then also be used to, e.g., evaluate the role of stochasticity, which the introduction hints might be important (but which then seems to vanish in the methods? but I'm a bit unclear on the methods, as detailed below, so might be wrong). Layering in a two sex model seems to me to be the main innovation - but then it would be good to know what the effects of asex vs. sexual host populations might be?

As the manuscript stands, the methods are a bit mystifying. Where is β defined? Where is α defined? What does the e subscript stand for? Could we get some sort of life cycle graph to help maybe with arrows aligned with the relevant parameters? Are the mating parameters supposed to be a ratio somehow? Or is that apparent dividing line a typo? Could you possibly reframe the mating function using classic forms (i.e., the 'marriage' function e.g., introduced via references available here? <https://link.springer.com/article/10.1007/s10144-018-0615-8>). (Or maybe this does fall within this framework, but I'm having trouble parsing it?). Sexual reproduction comes with seas of complexities (e.g., see <https://www.sciencedirect.com/science/article/pii/S0040580918302028> for protected polymorphisms, etc) that presumably translate into microbe effects too?

Is there any density dependence operating on hosts (e.g., in offspring establishment?) Frequency dependence? Might these matter? Presumably they could affect the stability of associations / necessity for environmental transmission on top of vertical transmission? If we're throwing stochasticity in as well, then presumably issues associated with non-linear averaging might also emerge, which could be very interesting? It would be great to have clearly annotated code as part of this manuscript.

Although I'm a little shaky on what the methods are doing in general, it does seem clear that, relative to previous work (see refs below), here, 'environmental' transmission is divorced from

dynamics across other hosts - there is simply a probability of acquisition of the microbe from the environment, or not. The nice broad overview of modes of transmission early in the manuscript makes a reasonably strong case that this might be expected (e.g., in the case of ingested microbes from the environment) but a more complete treatment could also include more usually treated flavours of transmission, including horizontal transmission as more usually framed, where the rate of acquisition depends on the the number of other hosts 'infected'. This type of horizontal transmission will bring with it a range of variance depending on the expected magnitude of the 'force of infection' or rate at which susceptible individuals become infected (lower force of infection = greater variance in age of acquisition, etc) which might be associated with a lot of interesting outcomes.

Finally, I think the results might be usefully placed within a broader modeling literature. There are a few further references that it might be helpful to include (listed below; two by Roughgarden, one by Vliet and Doebeli). These focus on the role of a multi-species microbiome (rather than a single species), but I think provide helpful perspective. The classic paper by Lipsitch et al. (as you indicate in the manuscript) showed that combining VT and HT may greatly increase the range of ecological conditions that support symbionts. It would be nice to more crisply frame what is added here. The neutral case? Environmental transmission? Sex differences? The role of a background horizontal rate? Environmental transmission seems at first glance as if the predictions might be rather obvious - higher levels = simply more individuals 'infected' - so what are the surprising counter-intuitive outcomes? Is it to do with the interaction with vertical transmission? From memory, the Lipsitch paper cited does a fantastic job of teasing apart how these things are operating and might be a good model for your framing of results? There is also a very useful Lipsitch et al. on evolution of virulence in the context of HT and VT published in 1996 which speaks to your case where the microbe decreases fitness.

Specific points

Perhaps the title could be changed to something more specific, depending on direction the paper takes?

L33 What 3 key questions? It would be helpful to specify?

L85 - ultimate determinants? proximate determinants?

L88/89 - so we are focussed on animal hosts then? perhaps be explicit about this in the title / introduction?

L149 - "assembled stochastically or deterministically" - should this be tested in the model? you make a strong case that it matters?

L237 - word missing?

L262 - is the word symbiont appropriate since also considering the negative case?

L279 - it is hard for me to figure out what is going on from the model, but if the fitness effects of the symbiont were zero, surely it isn't surprising that the fraction with microbes declines if only a subset of this fraction is conveyed by the mother to the next generation? it would be like it having $R_0 < 1$

L290 - this set up is rather odd - why not simply have one patch with an introduction rate? or something along the lines of more classic invasibility analysis? or is there a realistic scenario that we are trying to reflect?

L300 - required for what?

L365 - 'very modest' - relative to what?

References

Roughgarden et al. 2018 Holobionts as units of selection and a model of their population dynamics and evolution. *Biological Theory* 13 44-65

Roughgarden 2018 Model for vertical vs. horizontal microbial colonization bioRxiv:465310

Vliet & Doebeli 2019 The role of multilevel selection in host microbiome evolution PNAS

Review form: Reviewer 2

Recommendation

Major revision is needed (please make suggestions in comments)

Scientific importance: Is the manuscript an original and important contribution to its field?

Excellent

General interest: Is the paper of sufficient general interest?

Excellent

Quality of the paper: Is the overall quality of the paper suitable?

Acceptable

Is the length of the paper justified?

Yes

Should the paper be seen by a specialist statistical reviewer?

Yes

Do you have any concerns about statistical analyses in this paper? If so, please specify them explicitly in your report.

Yes

It is a condition of publication that authors make their supporting data, code and materials available - either as supplementary material or hosted in an external repository. Please rate, if applicable, the supporting data on the following criteria.

Is it accessible?

Yes

Is it clear?

No

Is it adequate?

No

Do you have any ethical concerns with this paper?

No

Comments to the Author

Overall this is a very interesting contribution to symbiosis research. The writing was clear and the background literature was well presented. That mixed mode transmission may be critical for maintenance of symbioses regardless of fitness impacts could be a very important finding, with implications in microbiome, epidemiology, vectored diseases, evolution, and ecology. However, the manuscript is not written for this broader audience and it should be revised to help non-experts understand both how the findings were obtained and their implications, as well as recommendations for how experimental systems could be used to test the predictions of the model.

The major criticism I have of the manuscripts pertains to one of the primary, and potentially most important conclusions drawn by the authors: that fitness plays only a “modest” role if any in symbiont transmission. My criticism stems from the fact that the authors seem to downplay the fitness effects that do appear in their models, without sufficient justification or rationale for categorizing this as “modest”. For example, the authors chose to separate from each other the four fitness levels tested (side note – the authors should define epsilon and these arbitrary fitness levels for the reader). Fitness levels of 0.75 and 1.33 are shown in one figure, and 1.0 and 2.0 are shown in another. If one examines all four side by side, in increasing order of fitness, a pattern does emerge. Examining only the 0.7 steady state carriage, the intersection of the line with vertical transmission frequency decreases as the fitness increases, and a similar trend occurs on the environmental transmission axis. These trends are consistent with the idea that fitness is important and can counter imperfect transmission. Put another way, the model predicts that a system with higher fitness would require slightly less probability of transmission by either environmental or maternal routes to achieve the same carriage. The difference is slight, but maybe meaningful in an evolutionary context. I think the authors need to do a better job of convincing the reader that this relationship is not significant. It would be interesting to plot the relationship between fitness and steady state carriage reliance on one or the other type of transmission and calculate the slope of that relationship.

On a related point, the wording for the requirement for vertical transmission in inter-population transmission is confusing, and possibly inaccurate. The graphs show that as long as fitness benefits are high (2.0) the perfect vertical transmission is not necessary, but the wording in both the text (L338-340) and the figure legend title for Fig. 2 both indicate that vertical transmission is a pre-requisite to steady state carriage and that benefits only slightly impact it. However, the interpretation is the other way around: benefits can compensate for less-than-perfect transmission. Hopefully that’s clear – apologies if not.

A second major concern I have is that the model does not have any verification. What did the authors do to ensure that their formula calculations were robust and not unduly impacted by incorrect assumptions? Are there any existing data that could be plotted to determine if they fit the model? Perhaps aphids, for which there is a wealth of knowledge from both lab and field studies and for which mixed mode transmission has been demonstrated?

On a more granular level, the way the data were presented were difficult to understand and interpret. Of note:

- The supplemental needed more information on how the equations were derived so that an expert could re-evaluate them.
- Explanation for the degrees of mate choice not described clearly enough in either the text or in the supplemental.
- The mate choice information is not clear. Are the panels in Fig. 1 categorized by male mate choice preference, with the degrees of freedom (1-9) indicating female sampling? I think so from the supplemental but it’s not clear enough.
- Individual panels are dense with information and it becomes difficult to tease apart meaningful from meaningless trends. For example, for the 0.5 steady state carriage line in Figs. 1A and 1B there is only one line apparent (rather than the multiples that are representing all mate choice degrees of freedom). For part of the line (low end of the vertical transmission axis) the

lightest shading (equating with 9 of the degrees of mate choice) shows, and presumably all the other lines track with this one. However as soon as the line crosses the vertical transmission frequency of 0.8 it becomes a dark line. Either this is a formatting issue or it has some meaning. Either it has to be fixed or explained.

- Supplemental: What are alpha and beta? These may be standard in math models, but for the general audience it will not be clear.
- In Figure 2, when population 1 is the only source of symbionts for transmission to population 2, what is the assumed steady state carriage in population 1?
- Iterative calculation should be explained. I think what they mean is that they have a starting carriage level and at that level determining the assortative mating impact, then back and forth until they reach steady state. But this should not be left up to the reader to try and interpret. How many iterations does it take for the population to reach steady state and did this vary depending on the assumptions/parameters?

And one last minor comment:

In Figure 2 the $n=3$ and 7 are missing from the figure caption, even though they're listed in the legend.

Decision letter (RSPB-2019-2052.R0)

30-Oct-2019

Dear Dr Chapman:

I am writing to inform you that your manuscript RSPB-2019-2052 entitled "Transmission efficiency is a primary driver of host-microbe associations" has, in its current form, been rejected for publication in Proceedings B.

This action has been taken on the advice of referees, who have recommended that very substantial revisions are necessary. With this in mind we would be happy to consider a resubmission, provided the comments of the referees are fully addressed. However please note that this is not a provisional acceptance.

To upload a resubmitted manuscript, log into <http://mc.manuscriptcentral.com/prsb> and enter your Author Centre, where you will find your manuscript title listed under "Manuscripts with

Decisions." Under "Actions," click on "Create a Resubmission." Please be sure to indicate in your cover letter that it is a resubmission, and supply the previous reference number.

Sincerely,
 Professor Hans Heesterbeek
 mailto: proceedingsb@royalsociety.org

Associate Editor
 Board Member: 1
 Comments to Author:

Thank you for your submission to the special issue. Your work has now been reviewed by myself and two reviewers, and we all see great value in both the approaches you are taking and the question you are tackling. That said, both reviewers had substantial concerns about the state of the manuscript and the level of contribution. As such, we would like to invite you to resubmit the manuscript if you feel you can address the comments - many of which call for significant further work (including expanding the range of contexts considered), better description of the model and methods used, model verification, and a rethink of the conclusion that fitness plays only a "modest" role in transmission. Both reviewers have offered substantial, thoughtful, and thorough comments throughout that need to be addressed, and should the authors choose to address these and resubmit, there is also a chance that additional/new reviewers would be found, so I would ask them to think broadly about the revision. In the event that the authors feel they can expand their model and findings to satisfy the requests from reviewers, I look forward to receiving a resubmission, which I do believe has the potential to make a strong contribution to the literature.

Reviewer(s)' Comments to Author:
 Referee: 1
 Comments to the Author(s)

This paper contrasts different forms of transmission and explores their roles in driving host-microbe associations to calibrate the role of environmental acquisition vs. vertical transmission in shaping observed associations at the scale of a single microbe / host. While I do think this effort (or something like it) is potentially of value, I don't think that the current model framework or manuscript achieves this. Perhaps more carefully addressing and expanding the range of contexts considered would be of greater value? For example one could build up a hierarchy of frames from asexual hosts, to sexual, to sexual with assortative mating, to sexual with assortative mating and spatial aggregation for example. This could also be done in a relatively methodologically consistent way via a series of nested matrix models (with the largest scale capturing two or more spatial patches, see some of Caswell's recent model extensions). The framing could then also be used to, e.g., evaluate the role of stochasticity, which the introduction hints might be important (but which then seems to vanish in the methods? but I'm a bit unclear on the methods, as detailed below, so might be wrong). Layering in a two-sex model seems to me to be the main innovation - but then it would be good to know what the effects of asexual vs. sexual host populations might be?

As the manuscript stands, the methods are a bit mystifying. Where is β defined? Where is α defined? What does the e subscript stand for? Could we get some sort of life cycle graph to help maybe with arrows aligned with the relevant parameters? Are the mating parameters supposed to be a ratio somehow? Or is that apparent dividing line a typo? Could you possibly reframe the mating function using classic forms (i.e., the 'marriage' function e.g., introduced via references available here? <https://link.springer.com/article/10.1007/s10144-018-0615-8>). (Or maybe this does fall within this framework, but I'm having trouble parsing it?). Sexual reproduction comes with seas of complexities (e.g., see <https://www.sciencedirect.com/science/article/pii/S0040580918302028> for protected polymorphisms, etc) that presumably translate into microbe effects too?

Is there any density dependence operating on hosts (e.g., in offspring establishment?) Frequency dependence? Might these matter? Presumably they could affect the stability of associations / necessity for environmental transmission on top of vertical transmission? If we're throwing stochasticity in as well, then presumably issues associated with non-linear averaging might also emerge, which could be very interesting? It would be great to have clearly annotated code as part of this manuscript.

Although I'm a little shaky on what the methods are doing in general, it does seem clear that, relative to previous work (see refs below), here, 'environmental' transmission is divorced from dynamics across other hosts - there is simply a probability of acquisition of the microbe from the environment, or not. The nice broad overview of modes of transmission early in the manuscript makes a reasonably strong case that this might be expected (e.g., in the case of ingested microbes from the environment) but a more complete treatment could also include more usually treated flavours of transmission, including horizontal transmission as more usually framed, where the rate of acquisition depends on the the number of other hosts 'infected'. This type of horizontal transmission will bring with it a range of variance depending on the expected magnitude of the 'force of infection' or rate at which susceptible individuals become infected (lower force of infection = greater variance in age of acquisition, etc) which might be associated with a lot of interesting outcomes.

Finally, I think the results might be usefully placed within a broader modeling literature. There are a few further references that it might be helpful to include (listed below; two by Roughgarden, one by Vliet and Doebeli). These focus on the role of a multi-species microbiome (rather than a single species), but I think provide helpful perspective. The classic paper by Lipsitch et al. (as you indicate in the manuscript) showed that combining VT and HT may greatly increase the range of ecological conditions that support symbionts. It would be nice to more crisply frame what is added here. The neutral case? Environmental transmission? Sex differences? The role of a background horizontal rate? Environmental transmission seems at first glance as if the predictions might be rather obvious - higher levels = simply more individuals 'infected' - so what are the surprising counter-intuitive outcomes? Is it to do with the interaction with vertical transmission? From memory, the Lipsitch paper cited does a fantastic job of teasing apart how these things are operating and might be a good model for your framing of results? There is also a very useful Lipsitch et al. on evolution of virulence in the context of HT and VT published in 1996 which speaks to your case where the microbe decreases fitness.

Specific points

Perhaps the title could be changed to something more specific, depending on direction the paper takes?

L33 What 3 key questions? It would be helpful to specify?

L85 - ultimate determinants? proximate determinants?

L88/89 - so we are focussed on animal hosts then? perhaps be explicit about this in the title / introduction?

L149 - "assembled stochastically or deterministically" - should this be tested in the model? you make a strong case that it matters?

L237 - word missing?

L262 - is the word symbiont appropriate since also considering the negative case?

L279 - it is hard for me to figure out what is going on from the model, but if the fitness effects of

the symbiont were zero, surely it isn't surprising that the fraction with microbes declines if only a subset of this fraction is conveyed by the mother to the next generation? it would be like it having $R_0 < 1$

L290 - this set up is rather odd - why not simply have one patch with an introduction rate? or something along the lines of more classic invasibility analysis? or is there a realistic scenario that we are trying to reflect?

L300 - required for what?

L365 - 'very modest' - relative to what?

References

Roughgarden et al. 2018 Holobionts as units of selection and a model of their population dynamics and evolution. *Biological Theory* 13 44-65

Roughgarden 2018 Model for vertical vs. horizontal microbial colonization bioRxiv:465310

Vliet & Doebeli 2019 The role of multilevel selection in host microbiome evolution PNAS

Referee: 2

Comments to the Author(s)

Overall this is a very interesting contribution to symbiosis research. The writing was clear and the background literature was well presented. That mixed mode transmission may be critical for maintenance of symbioses regardless of fitness impacts could be a very important finding, with implications in microbiome, epidemiology, vectored diseases, evolution, and ecology. However, the manuscript is not written for this broader audience and it should be revised to help non-experts understand both how the findings were obtained and their implications, as well as recommendations for how experimental systems could be used to test the predictions of the model.

The major criticism I have of the manuscripts pertains to one of the primary, and potentially most important conclusions drawn by the authors: that fitness plays only a "modest" role if any in symbiont transmission. My criticism stems from the fact that the authors seem to downplay the fitness effects that do appear in their models, without sufficient justification or rationale for categorizing this as "modest". For example, the authors chose to separate from each other the four fitness levels tested (side note - the authors should define epsilon and these arbitrary fitness levels for the reader). Fitness levels of 0.75 and 1.33 are shown in one figure, and 1.0 and 2.0 are shown in another. If one examines all four side by side, in increasing order of fitness, a pattern does emerge. Examining only the 0.7 steady state carriage, the intersection of the line with vertical transmission frequency decreases as the fitness increases, and a similar trend occurs on the environmental transmission axis. These trends are consistent with the idea that fitness is important and can counter imperfect transmission. Put another way, the model predicts that a system with higher fitness would require slightly less probability of transmission by either environmental or maternal routes to achieve the same carriage. The difference is slight, but maybe meaningful in an evolutionary context. I think the authors need to do a better job of convincing the reader that this relationship is not significant. It would be interesting to plot the relationship between fitness and steady state carriage reliance on one or the other type of transmission and calculate the slope of that relationship.

On a related point, the wording for the requirement for vertical transmission in inter-population

transmission is confusing, and possibly inaccurate. The graphs show that as long as fitness benefits are high (2.0) the perfect vertical transmission is not necessary, but the wording in both the text (L338-340) and the figure legend title for Fig. 2 both indicate that vertical transmission is a pre-requisite to steady state carriage and that benefits only slightly impact it. However, the interpretation is the other way around: benefits can compensate for less-than-perfect transmission. Hopefully that's clear – apologies if not.

A second major concern I have is that the model does not have any verification. What did the authors do to ensure that their formula calculations were robust and not unduly impacted by incorrect assumptions? Are there any existing data that could be plotted to determine if they fit the model? Perhaps aphids, for which there is a wealth of knowledge from both lab and field studies and for which mixed mode transmission has been demonstrated?

On a more granular level, the way the data were presented were difficult to understand and interpret. Of note:

- The supplemental needed more information on how the equations were derived so that an expert could re-evaluate them.
- Explanation for the degrees of mate choice not described clearly enough in either the text or in the supplemental.
- The mate choice information is not clear. Are the panels in Fig. 1 categorized by male mate choice preference, with the degrees of freedom (1-9) indicating female sampling? I think so from the supplemental but it's not clear enough.
- Individual panels are dense with information and it becomes difficult to tease apart meaningful from meaningless trends. For example, for the 0.5 steady state carriage line in Figs. 1A and 1B there is only one line apparent (rather than the multiples that are representing all mate choice degrees of freedom). For part of the line (low end of the vertical transmission axis) the lightest shading (equating with 9 of the degrees of mate choice) shows, and presumably all the other lines track with this one. However as soon as the line crosses the vertical transmission frequency of 0.8 it becomes a dark line. Either this is a formatting issue or it has some meaning. Either it has to be fixed or explained.
- Supplemental: What are alpha and beta? These may be standard in math models, but for the general audience it will not be clear.
- In Figure 2, when population 1 is the only source of symbionts for transmission to population 2, what is the assumed steady state carriage in population 1?
- Iterative calculation should be explained. I think what they mean is that they have a starting carriage level and at that level determining the assortative mating impact, then back and forth until they reach steady state. But this should not be left up to the reader to try and interpret. How many iterations does it take for the population to reach steady state and did this vary depending on the assumptions/parameters?

And one last minor comment:

In Figure 2 the n=3 and 7 are missing from the figure caption, even though they're listed in the legend.

Author's Response to Decision Letter for (RSPB-2019-2052.R0)

See Appendix A.

RSPB-2020-0820.R0

Review form: Reviewer 3

Recommendation

Major revision is needed (please make suggestions in comments)

Scientific importance: Is the manuscript an original and important contribution to its field?

Good

General interest: Is the paper of sufficient general interest?

Good

Quality of the paper: Is the overall quality of the paper suitable?

Marginal

Is the length of the paper justified?

Yes

Should the paper be seen by a specialist statistical reviewer?

No

Do you have any concerns about statistical analyses in this paper? If so, please specify them explicitly in your report.

No

It is a condition of publication that authors make their supporting data, code and materials available - either as supplementary material or hosted in an external repository. Please rate, if applicable, the supporting data on the following criteria.

Is it accessible?

Yes

Is it clear?

Yes

Is it adequate?

Yes

Do you have any ethical concerns with this paper?

No

Comments to the Author

The manuscript 'Transmission efficiency drives host-microbe associations' explores and discusses different factors that may play a role in the establishment of host-microbial associations, particularly focusing on modes of transmission. The authors first review the literature on vertical, horizontal and mixed modes of transmission, and then use a theoretical model to assess how combinations of VT and HT lead to host-microbe associations in a population of hosts. This is a very interesting topic and I agree with the authors that a simple theoretical model as used here has the potential to provide great insights in the evolution of host-microbe associations. In its current form, however, I feel that the manuscript largely fails to do so. See below my comments, ordered by section.

Introduction/review

I appreciated the introduction and review of transmission modes, and the authors did a nice job of summarizing a complicated field. It would be helpful to mention the review in the abstract. There are several places where being more specific about what their terms mean would be helpful. For example, in this paper, 'stability' is the measure that marks an evolutionary host-microbe association. However, the authors never quite define what exactly is meant by stability — is this on the level of a host population? Is it a single microbe? Or some facet/functionality of the whole microbial community? Does it imply some fitness benefit? Especially for the MMT, it seems important to be clear about whether MMT applies to just one microbe, or to the whole community: Is it like in *Drosophila*, where different microbial species have different transmission modes (VT: *Wolbachia*, HT: *Acetobacter* and *Lactobacillus*), or, is it something like the *Acetobacter thailandicus* example (line 183-186), where the same microbe can have both pseudo-VT and HT modes? Or can it include both? Defining 'stability' more clearly early in the manuscript would help establish the eco-evolutionary scale at which the insights from the model apply.

Methods/results

The paper does a poor job in explaining, presenting and visualizing the methods and results. Without reading the SI, it is essentially impossible to understand and interpret the results (and even after reading the SI, I'm still a bit puzzled). For instance, it is unclear in the main text how VT, HT and social transmission are defined in the model, while these concepts are central to all results. Not all details need to be given in the main text, but it would help to give the readers some idea about all variables and processes.

I am convinced that there must be a better way to visualize the results. I find the figures hard to digest, and not very appealing. At first glance, they basically all look the same, so it is very difficult to see which differences the authors want me to focus on (e.g. is there an important difference that I should notice between Fig. 1a and 1b?).

I encourage the authors to think about a better way to visualize their results, better guiding the reader to detect the important patterns. I suggest to add lines for which host variation is zero, i.e. those combinations for which carrier frequencies are either 0 or 1, as these seem particularly interesting. Maybe it helps to present the results as a heatmap? Instead of showing two arbitrary values for relative fitness, it would be useful to show how results change as a function of fitness. For instance, a graph showing the relation between relative fitness (x-axis) and the required VT/HT (y-axis) to obtain a carrier frequency of e.g. 0.1, 0.5, 1. I believe that there are more possibilities to 'summarize' some of the results that are now presented in many almost identical graphs (both in the main text and SI), and that such graphs will greatly help to see emerging general patterns. Perhaps it also helps to start with presenting the results of the 'core model', only considering the effects of VT and HT, and then stepwise add more complexity (reproductive mode, social transmission, dispersal etc.). Figure S1 is useful, I suggest adding such a figure that explains the modeling procedure to the main text. Please add labels i)-iii) to figure S1, now it is unclear what the caption refers to.

Another important point is the way HT (environmental acquisition) is incorporated in the model. The authors write in the SI that HT is defined as '...the probability that an individual will have acquired the microbe in the time period between their birth and their becoming sexually mature.' Both VT and HT thus give the probability that an individual becomes a carrier, either from its mother, or from the environment. Does this imply that VT and HT are essentially the same in the model (the only difference being that non-carriers cannot transmit it to their offspring, irrespective of the population-level VT)? In the introduction (L206), the authors write that '...MMT will reduce the strength and consistency of VT...'. I'm wondering if the presented model captures this, as VT and HT are not mutually exclusive. If I understand the modeling procedure correctly, once a microbe is vertically transmitted, a host will never lose it, for instance through horizontal acquisition of competing microbes. Instead, the two modes of transmission act in an additive way, providing two independent routes by which hosts can acquire (but never lose)

microbes. It is thus not surprising that increasing either of them, or both of them, all increase carrier frequencies. I am not convinced, though, that the correct conclusion is that mixed modes of transmission lead to more stable host-microbe associations (but rather: more faithful transmission, from VT and/or from HT, leads to more stable host-microbe associations)?

I think this also relates to the somewhat unexpected result that relative fitness has a relatively small impact on the frequency of carriers. The probability of getting the microbe (via either HT or VT) is fixed, so new carriers are being re-introduced every generation as determined by their HT/VT, no matter their fitness consequences. In natural populations, however, I would expect there to be selection acting on HT/VT directly (e.g. where hosts would evolve ways to avoid the exposure to certain microbes). Although I think it is perfectly fine that this paper focuses on equilibrium frequencies given fixed transmission patterns, instead of on the evolution of modes of transmission, I think this should be made clearer throughout the text, through more careful formulation (for instance on L392: 'We...found that increases to host fitness...had only minimal effect on promoting host-microbe relationships'. Is this a valid conclusion? Host-microbe relationships are not promoted, but selection simply cannot get rid of them). Defining different terms more precise (see comments above), will help.

Specific comments about the current figures:

- 1) Both in figure 1 and figure 2: please explain what additional degrees of host mate choice means i.e. is $n=1$ the choosiest and $n=9$ less choosy?
- 2) Figure 2: population 1 and 2 should be labelled on the figures.
- 3) No need to add a legend to each panel, but do add a title to the legend.

Discussion

The discussion could benefit from additional contextualization of the results. In its current form, the review and modeling part of the manuscript are largely disconnected, and I was hoping that the discussion would bring the two together. For example, the results from the migration model are quite interesting – where assortative mating can modulate the reinforcement or relaxation of reproductive isolation across populations (L320-328). The authors repeat these findings in the discussion (L401-403), but it would be interesting to compare these results to some of the examples discussed/cited in the review section. While I understand this would mostly be speculation, I think a little bit of connection would help the reader understand how to apply their theoretical results to natural systems. The authors suggest that future experiments could be used to test the modeling predictions (L382-386), but it remains unclear what kind of data/experiment one would need.

The discussion of fitness effects warrants some additional contextualization. For example, L371-373, the authors connect phenotypic and behavioral effects to ecological conditions that promote a high frequency of carriers (is this different from stability? And is this really 'promoting', see my comment above? Another place to where fuzziness impedes the reader). The model doesn't really incorporate differences in ecological conditions, unless this is to be implied by the different populations? Perhaps application to empirical data would be helpful to better explain what is meant here. As stated earlier, I think this study could be a valuable contribution to our understanding in host-microbe associations, but a little bit more context is needed in the discussion to apply the theoretical findings.

Review form: Reviewer 4

Recommendation

Accept with minor revision (please list in comments)

Scientific importance: Is the manuscript an original and important contribution to its field?
Excellent

General interest: Is the paper of sufficient general interest?
Excellent

Quality of the paper: Is the overall quality of the paper suitable?
Good

Is the length of the paper justified?
Yes

Should the paper be seen by a specialist statistical reviewer?
No

Do you have any concerns about statistical analyses in this paper? If so, please specify them explicitly in your report.
No

It is a condition of publication that authors make their supporting data, code and materials available - either as supplementary material or hosted in an external repository. Please rate, if applicable, the supporting data on the following criteria.

Is it accessible?
Yes

Is it clear?
Yes

Is it adequate?
Yes

Do you have any ethical concerns with this paper?
No

Comments to the Author

This manuscript studies the evolutionary stability of host-microbe associations focusing on the efficacy of three types of microbe transmission modes: environmental acquisition, vertical transmission and horizontal transmission. The authors found that mixed modes of transmission (MMT), mostly involving environmental acquisition and vertical transmission are more efficient in stabilising host-microbe associations across generations than each transmission mode alone. The authors review the literature of microbe transmission modes and discuss the relevance of their findings.

In my opinion, the manuscript provides an important contribution to the field, as it improves our understanding of the interplay between microbe transmission modes, with implications to reproductive isolation and the persistence of associations after dispersal. However, in my opinion, the manuscript still needs to be improved in some respects. Below I describe my major and minor concerns in detail:

Major concerns

The meaning of «stabilizing», «stable» and «persistent»

The authors use these words throughout the manuscript, but they are not explicit as to their meaning, which can be twofold: it may be the transition from transient to permanent infection at the individual level; but it can also be the persistence of infection over evolutionary time, that is, the transition from transient to permanent infection at the population level and across

generations. It may not be immediately obvious to the reader that these concepts refer to the persistence of infection at the population level. So, I advise the authors to make this clearer in the text, especially in the abstract and introduction, where the concepts appear for the first time.

The concepts of Environmental Acquisition (EA), Vertical Transmission (VT), Horizontal Transmission (HT) and Social Transmission (ST)

The manuscript is not coherent about the use of these concepts. First, ST appears only a few times in the main text but appears frequently in the supplementary material, including in all figures. Second, ST is combined in the main text with the other transmission modes, while in the supplementary material it is shown as an independent mode of transmission. These discrepancies make the manuscript confusing. If ST was tested as an independent transmission mode, it should appear in the main text as such. Besides, ST, VT, HT and EA are not synonyms. EA should not be called HT because it is not a transmission mode between social agents but between animals and their external (asocial) environment; ST, on the other hand, can be either vertical, horizontal or oblique if it is between parents and offspring (VT), older and younger individuals not directly related (OT) or between individuals of the same age class (HT). Because the models that were here developed simulate discrete generations, there is no OT and so this term can be omitted. However, for the remaining concepts, I advise the authors to use them in a more systematized way: use ST only as a general term for VT and HT and use VT, EA and HT in all other cases, including the model parameter and all the figures.

This also implies that the authors change the second section of the introduction (lines 91-165): this section should be only about EA, and the authors should include a new section only about HT. Finally, the section about MMT (lines 167-212) should include references to all three transmission modes: VT, EA and HT.

Evolutionary significance of host-microbe associations

I do not fully understand the evolutionary component of the mathematical models the authors developed. Although the models simulate steady-state microbe carrier frequencies, I suppose the populations reached those frequencies after several generations. But this is not visible in the figures and is also difficult to extract from the description of the models in the Supplementary Material (SM).

To improve the manuscript on this respect, the authors could provide the simulation algorithm, the sequence of steps since the creation of the simulated world, with a certain number of individuals of each type, until reproduction, migration, survival, etc. What kind of world it is: a grid with patches or a unique patch? What is the sequence of steps? How many individuals begin the simulations and how many survive each generation? How many generations? Unless there is a method section (and not just a supplementary method section) that I did not have access to, this information needs to be added to the manuscript.

Minor concerns

Lines 35-36: Here, the authors could mention that microbes can also BE SOCIALLY TRANSMITTED.

Line 57: At the end of that sentence, the authors could state they will also review «the as yet under-researched evolutionary potential for mixed modes of transmission».

Line 59: «Heritable» is redundant with «vertical transmission». Besides, microbes transmitted horizontally are also heritable.

Lines 88-89: Please, specify that coprophagy and social acquisition are between mother and offspring in that case. This is important because horizontal transmission is also a type of social transmission.

Line 115-116: From my understanding, this is not a case of EA but HT. This example should be in a separate section dedicated to HT.

Lines 100-107: These two sentences provide contradictory evidence about the co-occurrences between microbial communities and hosts. Although these are two possible scenarios, they are not logically interconnected and seem to contradict each other. I suggest pulling these two sentences in the same paragraph and interconnect the two ideas better.

Lines 107-117: Shouldn't this paragraph be at the beginning of section (ii)?

Lines 124-125: That sentence should have a reference, and it would be clearer if the authors explain how transient gut microbiota can cause RI.

Line 168: The authors could write «highly specialised INTRACELLULAR host-microbe associations». This will help to contrast with the gut microbiome system, that is not intracellular, in the next paragraph (line 179).

Lines 189-191: I do not understand that sentence. How can parent-offspring transmission occur if it is mediated by independent replication of microbes in the environment?

Line 220: «established residencies with hosts, which then spread at a population level» AND ACROSS GENERATIONS, right?

Lines 226-227: This is just a detail, but it would be better if the authors recapitulate the modes of transmission they considered for their models (VT, EA, HT and MMT) and then inform that they were tested either singly (VT, EA and HT) or in combination (MMT).

Lines 228-230: The authors simulated two populations to test reproductive isolation, but it is missing from the main questions/ goals. They could add that goal to the second question.

Lines 397-411: The results the authors obtained with social transmission (which they should call HT) were not discussed. I think a discussion of these results should be included here.

Titles of figures 1 and 2: The titles of the figures are very different from each other. The title of figure 2 is a result and that of figure 1 is a description of the model parameters. It would make more sense for both figures to have similar titles. My advice is that the titles are based on the description of the model parameters. And, in turn, the subtitles of section 2 (lines 253-255 and 307-309) are a description of the results (like in figure 2).

Legends of figures 1 and 2: The legend of the figures should include a brief explanation of the meaning of n in «the degrees of host mate choice (n= 1,3,5,6 or 9)».

Figures from the SM: The letters of the graph axes are too small.

Decision letter (RSPB-2020-0820.R0)

08-Jun-2020

Dear Dr Chapman:

Your manuscript has now been peer reviewed and the reviews have been assessed by an Associate Editor. The reviewers' comments (not including confidential comments to the Editor) and the comments from the Associate Editor are included at the end of this email for your reference. As you will see, the reviewers and the Editors have raised substantial concerns with your manuscript and we would like to invite you to revise your manuscript to address them.

Research ethics:

Use of animals and field studies:

Please submit a copy of your revised paper within three weeks. If we do not hear from you within this time your manuscript will be rejected. If you are unable to meet this deadline please let us know as soon as possible, as we may be able to grant a short extension.

Best wishes,
Professor Hans Heesterbeek
mailto:proceedingsb@royalsociety.org

Associate Editor Board Member

Comments to Author:

Thank you for submitting your work for consideration in the special issue. Your manuscript has now been read by myself and two reviewers, and while we all see value in the work and a great need for these types of analyses, both reviewers found parts of the manuscript (especially the presentation and visualization of the results) difficult to understand. Given that we all feel this could make a strong contribution to the literature, I would like to ask the authors to revise the work in light of these thoughtful and thorough comments. I look forward to reading the revised manuscript, and thank you in advance for your work.

Reviewer(s)' Comments to Author:

Referee: 3

Comments to the Author(s).

The manuscript 'Transmission efficiency drives host-microbe associations' explores and discusses different factors that may play a role in the establishment of host-microbial associations, particularly focusing on modes of transmission. The authors first review the literature on vertical, horizontal and mixed modes of transmission, and then use a theoretical model to assess how combinations of VT and HT lead to host-microbe associations in a population of hosts. This is a very interesting topic and I agree with the authors that a simple theoretical model as used here has the potential to provide great insights in the evolution of host-microbe associations. In its current form, however, I feel that the manuscript largely fails to do so. See below my comments, ordered by section.

Introduction/review

I appreciated the introduction and review of transmission modes, and the authors did a nice job of summarizing a complicated field. It would be helpful to mention the review in the abstract. There are several places where being more specific about what their terms mean would be helpful. For

example, in this paper, 'stability' is the measure that marks an evolutionary host-microbe association. However, the authors never quite define what exactly is meant by stability – is this on the level of a host population? Is it a single microbe? Or some facet/functionality of the whole microbial community? Does it imply some fitness benefit? Especially for the MMT, it seems important to be clear about whether MMT applies to just one microbe, or to the whole community: Is it like in *Drosophila*, where different microbial species have different transmission modes (VT: *Wolbachia*, HT: *Acetobacter* and *Lactobacillus*), or, is it something like the *Acetobacter thailandicus* example (line 183-186), where the same microbe can have both pseudo-VT and HT modes? Or can it include both? Defining 'stability' more clearly early in the manuscript would help establish the eco-evolutionary scale at which the insights from the model apply.

Methods/results

The paper does a poor job in explaining, presenting and visualizing the methods and results. Without reading the SI, it is essentially impossible to understand and interpret the results (and even after reading the SI, I'm still a bit puzzled). For instance, it is unclear in the main text how VT, HT and social transmission are defined in the model, while these concepts are central to all results. Not all details need to be given in the main text, but it would help to give the readers some idea about all variables and processes.

I am convinced that there must be a better way to visualize the results. I find the figures hard to digest, and not very appealing. At first glance, they basically all look the same, so it is very difficult to see which differences the authors want me focus on (e.g. is there an important difference that I should notice between Fig. 1a and 1b?).

I encourage the authors to think about a better way to visualize their results, better guiding the reader to detect the important patterns. I suggest to add lines for which host variation is zero, i.e. those combinations for which carrier frequencies are either 0 or 1, as these seem particularly interesting. Maybe it helps to present the results as a heatmap? Instead of showing two arbitrary values for relative fitness, it would be useful to show how results change as a function of fitness. For instance, a graph showing the relation between relative fitness (x-axis) and the required VT/HT (y-axis) to obtain a carrier frequency of e.g. 0.1, 0.5, 1. I believe that there are more possibilities to 'summarize' some of the results that are now presented in many almost identical graphs (both in the main text and SI), and that such graphs will greatly help to see emerging general patterns. Perhaps it also helps to start with presenting the results of the 'core model', only considering the effects of VT and HT, and then stepwise add more complexity (reproductive mode, social transmission, dispersal etc.). Figure S1 is useful, I suggest adding such a figure that explains the modeling procedure to the main text. Please add labels i)-iii) to figure S1, now it is unclear what the caption refers to.

Another important point is the way HT (environmental acquisition) is incorporated in the model. The authors write in the SI that HT is defined as '...the probability that an individual will have acquired the microbe in the time period between their birth and their becoming sexually mature.' Both VT and HT thus give the probability that an individual becomes a carrier, either from its mother, or from the environment. Does this imply that VT and HT are essentially the same in the model (the only difference being that non-carriers cannot transmit it to their offspring, irrespective of the population-level VT)? In the introduction (L206), the authors write that '...MMT will reduce the strength and consistency of VT...'. I'm wondering if the presented model captures this, as VT and HT are not mutually exclusive. If I understand the modeling procedure correctly, once a microbe is vertically transmitted, a host will never lose it, for instance through horizontal acquisition of competing microbes. Instead, the two modes of transmission act in an additive way, providing two independent routes by which hosts can acquire (but never lose) microbes. It is thus not surprising that increasing either of them, or both of them, all increase carrier frequencies. I am not convinced, though, that the correct conclusion is that mixed modes of transmission lead to more stable host-microbe associations (but rather: more faithful transmission, from VT and/or from HT, leads to more stable host-microbe associations)?

I think this also relates to the somewhat unexpected result that relative fitness has a relatively small impact on the frequency of carriers. The probability of getting the microbe (via either HT or VT) is fixed, so new carriers are being re-introduced every generation as determined by their HT/VT, no matter their fitness consequences. In natural populations, however, I would expect there to be selection acting on HT/VT directly (e.g. where hosts would evolve ways to avoid the exposure to certain microbes). Although I think it is perfectly fine that this paper focuses on equilibrium frequencies given fixed transmission patterns, instead of on the evolution of modes of transmission, I think this should be made clearer throughout the text, through more careful formulation (for instance on L392: 'We...found that increases to host fitness...had only minimal effect on promoting host-microbe relationships'. Is this a valid conclusion? Host-microbe relationships are not promoted, but selection simply cannot get rid of them). Defining different terms more precise (see comments above), will help.

Specific comments about the current figures:

- 1) Both in figure 1 and figure 2: please explain what additional degrees of host mate choice means i.e. is $n=1$ the choosiest and $n=9$ less choosy?
- 2) Figure 2: population 1 and 2 should be labelled on the figures.
- 3) No need to add a legend to each panel, but do add a title to the legend.

Discussion

The discussion could benefit from additional contextualization of the results. In its current form, the review and modeling part of the manuscript are largely disconnected, and I was hoping that the discussion would bring the two together. For example, the results from the migration model are quite interesting – where assortative mating can modulate the reinforcement or relaxation of reproductive isolation across populations (L320-328). The authors repeat these findings in the discussion (L401-403), but it would be interesting to compare these results to some of the examples discussed/cited in the review section. While I understand this would mostly be speculation, I think a little bit of connection would help the reader understand how to apply their theoretical results to natural systems. The authors suggest that future experiments could be used to test the modeling predictions (L382-386), but it remains unclear what kind of data/experiment one would need.

The discussion of fitness effects warrants some additional contextualization. For example, L371-373, the authors connect phenotypic and behavioral effects to ecological conditions that promote a high frequency of carriers (is this different from stability? And is this really 'promoting', see my comment above? Another place to where fuzziness impedes the reader). The model doesn't really incorporate differences in ecological conditions, unless this is to be implied by the different populations? Perhaps application to empirical data would be helpful to better explain what is meant here. As stated earlier, I think this study could be a valuable contribution to our understanding in host-microbe associations, but a little bit more context is needed in the discussion to apply the theoretical findings.

Referee: 4

Comments to the Author(s).

This manuscript studies the evolutionary stability of host-microbe associations focusing on the efficacy of three types of microbe transmission modes: environmental acquisition, vertical transmission and horizontal transmission. The authors found that mixed modes of transmission (MMT), mostly involving environmental acquisition and vertical transmission are more efficient in stabilising host-microbe associations across generations than each transmission mode alone. The authors review the literature of microbe transmission modes and discuss the relevance of their findings.

In my opinion, the manuscript provides an important contribution to the field, as it improves our understanding of the interplay between microbe transmission modes, with implications to reproductive isolation and the persistence of associations after dispersal. However, in my opinion, the manuscript still needs to be improved in some respects. Below I describe my major and minor concerns in detail:

Major concerns

The meaning of «stabilizing», «stable» and «persistent»

The authors use these words throughout the manuscript, but they are not explicit as to their meaning, which can be twofold: it may be the transition from transient to permanent infection at the individual level; but it can also be the persistence of infection over evolutionary time, that is, the transition from transient to permanent infection at the population level and across generations. It may not be immediately obvious to the reader that these concepts refer to the persistence of infection at the population level. So, I advise the authors to make this clearer in the text, especially in the abstract and introduction, where the concepts appear for the first time.

The concepts of Environmental Acquisition (EA), Vertical Transmission (VT), Horizontal Transmission (HT) and Social Transmission (ST)

The manuscript is not coherent about the use of these concepts. First, ST appears only a few times in the main text but appears frequently in the supplementary material, including in all figures. Second, ST is combined in the main text with the other transmission modes, while in the supplementary material it is shown as an independent mode of transmission. These discrepancies make the manuscript confusing. If ST was tested as an independent transmission mode, it should appear in the main text as such. Besides, ST, VT, HT and EA are not synonyms. EA should not be called HT because it is not a transmission mode between social agents but between animals and their external (asocial) environment; ST, on the other hand, can be either vertical, horizontal or oblique if it is between parents and offspring (VT), older and younger individuals not directly related (OT) or between individuals of the same age class (HT). Because the models that were here developed simulate discrete generations, there is no OT and so this term can be omitted. However, for the remaining concepts, I advise the authors to use them in a more systematized way: use ST only as a general term for VT and HT and use VT, EA and HT in all other cases, including the model parameter and all the figures.

This also implies that the authors change the second section of the introduction (lines 91-165): this section should be only about EA, and the authors should include a new section only about HT. Finally, the section about MMT (lines 167-212) should include references to all three transmission modes: VT, EA and HT.

Evolutionary significance of host-microbe associations

I do not fully understand the evolutionary component of the mathematical models the authors developed. Although the models simulate steady-state microbe carrier frequencies, I suppose the populations reached those frequencies after several generations. But this is not visible in the figures and is also difficult to extract from the description of the models in the Supplementary Material (SM).

To improve the manuscript on this respect, the authors could provide the simulation algorithm, the sequence of steps since the creation of the simulated world, with a certain number of individuals of each type, until reproduction, migration, survival, etc. What kind of world it is: a grid with patches or a unique patch? What is the sequence of steps? How many individuals begin the simulations and how many survive each generation? How many generations? Unless there is a method section (and not just a supplementary method section) that I did not have access to, this information needs to be added to the manuscript.

Minor concerns

Lines 35-36: Here, the authors could mention that microbes can also BE SOCIALLY TRANSMITTED.

Line 57: At the end of that sentence, the authors could state they will also review «the as yet under-researched evolutionary potential for mixed modes of transmission».

Line 59: «Heritable» is redundant with «vertical transmission». Besides, microbes transmitted horizontally are also heritable.

Lines 88-89: Please, specify that coprophagy and social acquisition are between mother and offspring in that case. This is important because horizontal transmission is also a type of social transmission.

Line 115-116: From my understanding, this is not a case of EA but HT. This example should be in a separate section dedicated to HT.

Lines 100-107: These two sentences provide contradictory evidence about the co-occurrences between microbial communities and hosts. Although these are two possible scenarios, they are not logically interconnected and seem to contradict each other. I suggest pulling these two sentences in the same paragraph and interconnect the two ideas better.

Lines 107-117: Shouldn't this paragraph be at the beginning of section (ii)?

Lines 124-125: That sentence should have a reference, and it would be clearer if the authors explain how transient gut microbiota can cause RI.

Line 168: The authors could write «highly specialised INTRACELLULAR host-microbe associations». This will help to contrast with the gut microbiome system, that is not intracellular, in the next paragraph (line 179).

Lines 189-191: I do not understand that sentence. How can parent-offspring transmission occur if it is mediated by independent replication of microbes in the environment?

Line 220: «established residencies with hosts, which then spread at a population level» AND ACROSS GENERATIONS, right?

Lines 226-227: This is just a detail, but it would be better if the authors recapitulate the modes of transmission they considered for their models (VT, EA, HT and MMT) and then inform that they were tested either singly (VT, EA and HT) or in combination (MMT).

Lines 228-230: The authors simulated two populations to test reproductive isolation, but it is missing from the main questions/goals. They could add that goal to the second question.

Lines 397-411: The results the authors obtained with social transmission (which they should call HT) were not discussed. I think a discussion of these results should be included here.

Titles of figures 1 and 2: The titles of the figures are very different from each other. The title of figure 2 is a result and that of figure 1 is a description of the model parameters. It would make more sense for both figures to have similar titles. My advice is that the titles are based on the description of the model parameters. And, in turn, the subtitles of section 2 (lines 253-255 and 307-309) are a description of the results (like in figure 2).

Legends of figures 1 and 2: The legend of the figures should include a brief explanation of the meaning of n in «the degrees of host mate choice (n= 1,3,5,6 or 9)».

Figures from the SM: The letters of the graph axes are too small.

Author's Response to Decision Letter for (RSPB-2020-0820.R0)

See Appendix B.

Decision letter (RSPB-2020-0820.R1)

05-Aug-2020

Dear Dr Chapman

I am pleased to inform you that your Review manuscript RSPB-2020-0820.R1 entitled "Transmission efficiency drives host-microbe associations" has been accepted for publication in Proceedings B.

The referee(s) do not recommend any further changes. Therefore, please proof-read your manuscript carefully and upload your final files for publication. Because the schedule for publication is very tight, it is a condition of publication that you submit the revised version of your manuscript within 7 days. If you do not think you will be able to meet this date please let me know immediately.

To upload your manuscript, log into <http://mc.manuscriptcentral.com/prsb> and enter your Author Centre, where you will find your manuscript title listed under "Manuscripts with Decisions." Under "Actions," click on "Create a Revision." Your manuscript number has been appended to denote a revision.

You will be unable to make your revisions on the originally submitted version of the manuscript. Instead, upload a new version through your Author Centre.

- 1) A text file of the manuscript (doc, txt, rtf or tex), including the references, tables (including captions) and figure captions. Please remove any tracked changes from the text before submission. PDF files are not an accepted format for the "Main Document".
- 2) A separate electronic file of each figure (tiff, EPS or print-quality PDF preferred). The format should be produced directly from original creation package, or original software format. Please note that PowerPoint files are not accepted.
- 3) Electronic supplementary material: this should be contained in a separate file from the main text and the file name should contain the author's name and journal name, e.g. `authorname_procb_ESM_figures.pdf`
All supplementary materials accompanying an accepted article will be treated as in their final form. They will be published alongside the paper on the journal website and posted on the online figshare repository. Files on figshare will be made available approximately one week before the accompanying article so that the supplementary material can be attributed a unique DOI. Please see: <https://royalsociety.org/journals/authors/author-guidelines/>

4) Data-Sharing and data citation

It is a condition of publication that data supporting your paper are made available. Data should be made available either in the electronic supplementary material or through an appropriate repository. Details of how to access data should be included in your paper. Please see <https://royalsociety.org/journals/ethics-policies/data-sharing-mining/> for more details.

If you wish to submit your data to Dryad (<http://datadryad.org/>) and have not already done so you can submit your data via this link <http://datadryad.org/submit?journalID=RSPB&manu=RSPB-2020-0820.R1> which will take you to your unique entry in the Dryad repository.

Once again, thank you for submitting your manuscript to Proceedings B and I look forward to receiving your final version. If you have any questions at all, please do not hesitate to get in touch.

Sincerely,
Dr The Proceedings B Team
Editor, Proceedings B
<mailto:proceedingsb@royalsociety.org>

Associate Editor Board Member

Comments to Author:

Thank you for taking the time to so carefully and thoughtfully respond to reviewer comments/suggestions. I think the work will make an excellent contribution to the special issue.

Decision letter (RSPB-2020-0820.R2)

10-Aug-2020

Dear Dr Chapman

I am pleased to inform you that your manuscript entitled "Transmission efficiency drives host-microbe associations" has been accepted for publication in Proceedings B.

Open Access

Corresponding authors from member institutions (<http://royalsocietypublishing.org/site/librarians/allmembers.xhtml>) receive a 25% discount to these charges. For more information please visit <http://royalsocietypublishing.org/open-access>.

Paper charges

Sincerely,
Proceedings B
<mailto:proceedingsb@royalsociety.org>

Dear colleagues,

Many thanks for the opportunity to resubmit a revised version of this MS. We really appreciated the extensive and constructive comments provided by the editor and 2 reviewers. In response, we have conducted additional modelling, provide an extensive new supplement including a modelling strategy figure, and we have rewritten, tightened and restructured the MS throughout. We provide a point by point description of all the changes, below, as well as a highlighted version of the MS to indicate the major revisions. We believe that following the suggestions made has greatly improved the MS and we hope that it is now suitable for publication.

Very best wishes,
Tracey Chapman (on behalf of all authors)

Editor:

Thank you for your submission to the special issue. Your work has now been reviewed by myself and two reviewers, and we all see great value in both the approaches you are taking and the question you are tackling. That said, both reviewers had substantial concerns about the state of the manuscript and the level of contribution. As such, we would like to invite you to resubmit the manuscript if you feel you can address the comments - many of which call for significant further work (including expanding the range of contexts considered), better description of the model and methods used, model verification, and a rethink of the conclusion that fitness plays only a "modest" role in transmission. Both reviewers have offered substantial, thoughtful, and thorough comments throughout that need to be addressed, and should the authors choose to address these and resubmit, there is also a chance that additional/new reviewers would be found, so I would ask them to think broadly about the revision. In the event that the authors feel they can expand their model and findings to satisfy the requests from reviewers, I look forward to receiving a resubmission, which I do believe has the potential to make a strong contribution to the literature.

Thank you for this assessment, we have taken seriously your recommendation for the need to expand the range of contexts considered, to extend and provide a better description of the models, consider model verification and the relative importance of fitness in transmission. The detailed description of the changes are provided below and highlighted in the main MS.

Referee: 1

This paper contrasts different forms of transmission and explores their roles in driving host microbe associations to calibrate the role of environmental acquisition vs. vertical transmission in shaping observed associations at the scale of a single microbe / host. While I do think this effort (or something like it) is potentially of value, I don't think that the current model framework or manuscript achieves this. Perhaps more carefully addressing and expanding the range of contexts considered would be of greater value? For example one could build up a hierarchy of frames from asexual hosts, to sexual, to sexual with assortative mating, to sexual with assortative mating, and spatial aggregation for example.

Response 1: Thanks for this comment and for the suggestion of the frames, which adopted. We have clarified our description of our model frameworks to clearly state that we tested our model under a range of different sexual mating pressures - to include sexual mating spanning the range of disassortative and assortative mating. In addition, we have included a descriptor that runs through these frames in the hierarchical manner suggested by the reviewer. Lines 234-251; figure S1 + caption.

Response 2: In response to the reviewer's suggestions we have also included a new framework of asexual host dynamics - the full results of which are given in the supplementary materials (Fig S2A, B). Though we agree it was interesting to make this addition, we find that these results do not have a substantive impact on our findings. Lines 262-263; 318-319; 353-354.

This could also be done in a relatively methodologically consistent way via a series of nested matrix models (with the largest scale capturing two or more spatial patches, see some of Caswell's recent model extensions). The framing could then also be used to, e.g., evaluate the role of stochasticity, which the introduction hints

might be important (but which then seems to vanish in the methods? but I'm a bit unclear on the methods, as detailed below, so might be wrong). Layering in a two sex model seems to me to be the main innovation - but then it would be good to know what the effects of asex vs. sexual host populations might be?

Response 3: Thanks for this suggestion to layer in sex vs asex populations. We followed this recommendation and conducted additional modelling to examine this specifically. Lines 228-230; 236-237; 262-263; 318-319; 353-354; Figs S2A, B.

Response 4: regarding the suggestion to evaluate the role of stochasticity, we restricted our approach to deterministic modelling due to the number of distinct modes of microbe transmission we wished to consider. The advantage of this was that it allowed us to easily disentangle the impacts of each parameter within our model - an issue that can often be challenging when using stochastic methods. We also anticipate that, in considering a sufficiently large population, stochastic effects would only play a significant role when microbe frequencies are very low. Given we are seeking to explore mechanisms leading a microbe to spread to high frequency, we would not anticipate this issue to have a significant or biasing impact on results we present here. To register and acknowledge this point, we added a discussion to the MS on lines 413-420.

As the manuscript stands, the methods are a bit mystifying. Where is beta defined? Where is alpha defined? What does the e subscript stand for? Could we get some sort of life cycle graph to help maybe with arrows aligned with the relevant parameters? Are the mating parameters supposed to be a ratio somehow? Or is that apparent dividing line a typo? Could you possibly reframe the mating function using classic forms (i.e., the 'marriage' function e.g., introduced via references available here?

Response 5: We are sorry that this was a challenge to unpick, that was not our intention! To address this point, we have now included definitions of all mathematical terms in a fully expanded supplementary methods section. The addition of a life cycle graph is an excellent idea and we followed this by introducing a new figure in the paper (figure 1). We have also expanded our supplementary information to include in full detail all of the modelling parameters. To further explain the approach, we also cite in the SI Newberry MG, McCandlish DM, Plotkin JB. 2016 Assortative mating can impede or facilitate fixation of underdominant alleles. *Theor. Popul. Biol.* 112, 14-21. (doi:10.1016/j.tpb.2016.07.003) as the exemplar framework we used for building our mate choice function.

(Or maybe this does fall within this framework, but I'm having trouble parsing it?). Sexual reproduction comes with a sea of complexities that presumably translate into microbe effects too?

Is there any density dependence operating on hosts (e.g., in offspring establishment?) Frequency dependence?

Response 6: Interesting, however, we did not include density dependent effects, as it would fundamentally change our modelling design. We do agree that it is potentially important, however, and now include a new section on how and why this would be a worthwhile inclusion for future work. Lines 420-424.

Might these matter? Presumably they could affect the stability of associations / necessity for environmental transmission on top of vertical transmission? If we're throwing stochasticity in as well, then presumably issues associated with non-linear averaging might also emerge, which could be very interesting? It would be great to have clearly annotated code as part of this manuscript.

Response 7: See also responses 3, 4, 6 where we clarify what factors we did include as part of the rationale for this current study, plus the extra new section, mentioned in response 6, where we highlight the likely influence and potential importance of these extra factors, and have added in this interesting point about plus the potential influence of non-linear averaging, lines 422-424. We have now included all the mathematical code in the SI and a link to the annotated Matlab code in the data accessibility statement (DOI 10.5281/zenodo.3746199).

Although I'm a little shaky on what the methods are doing in general, it does seem clear that, relative to previous work (see refs below), here, 'environmental' transmission is divorced from dynamics across other hosts - there is simply a probability of acquisition of the microbe from the environment, or not. The nice broad overview of modes of transmission early in the manuscript makes a reasonably strong case that this might be expected (e.g., in the case of ingested microbes from the environment) but a more complete

treatment could also include more usually treated flavours of transmission, including horizontal transmission as more usually framed, where the rate of acquisition depends on the number of other hosts 'infected'. This type of horizontal transmission will bring with it a range of variance depending on the expected magnitude of the 'force of infection' or rate at which susceptible individuals become infected (lower force of infection = greater variance in age of acquisition, etc) which might be associated with a lot of interesting outcomes.

Response 8: Agreed and thanks for this - we have now explored this more explicitly by including a rate of horizontal transmission - that is a probability based on the frequency of other infected individuals within the host population. Full details are included in the supplementary methods and figures and see also figure S1 caption.

Finally, I think the results might be usefully placed within a broader modeling literature. There are a few further references that it might be helpful to include (listed below; two by Roughgarden, one by Vliet and Doebeli). These focus on the role of a multi-species microbiome (rather than a single species), but I think provide helpful perspective. The classic paper by Lipsitch et al. (as you indicate in the manuscript) showed that combining VT and HT may greatly increase the range of ecological conditions that support symbionts. It would be nice to more crisply frame what is added here. The neutral case? Environmental transmission? Sex differences? The role of a background horizontal rate? Environmental transmission seems at first glance as if the predictions might be rather obvious - higher levels = simply more individuals 'infected' - so what are the surprising counter-intuitive outcomes? Is it to do with the interaction with vertical transmission? From memory, the Lipsitch paper cited does a fantastic job of teasing apart how these things are operating and might be a good model for your framing of results? There is also a very useful Lipsitch et al. on evolution of virulence in the context of HT and VT published in 1996 which speaks to your case where the microbe decreases fitness.

Response 9: Thanks for this and we agree that significant improvements to the framing of the results and their contribution was needed. To achieve this, we have substantially rewritten the discussion, added these important references and have fitted our modelling results into the framework they provide.

Specific points

Perhaps the title could be changed to something more specific, depending on direction the paper takes?
Done

L33 What 3 key questions? It would be helpful to specify?

Response 10: changed, **Lines 45-48; 107-117.**

L85 - ultimate determinants? proximate determinants?

Response 11: revised out.

L88/89 - so we are focussed on animal hosts then? perhaps be explicit about this in the title / introduction?

Response 12: yes – we have revised to make this clearer, **lines 40; 46; 53.**

L149 - "assembled stochastically or deterministically" - should this be tested in the model? you make a strong case that it matters?

Response 13: Changed to random - stochastic was incorrect, apologies - we wanted to pose the question as to whether there are functions involved for non-random assemblage or the microbiome or not. **Line 110.**

L237 - word missing?

Response 14: yes, corrected.

L262 - is the word symbiont appropriate since also considering the negative case?

Response 15: Our understanding is that symbiosis is a broad term to include parasites, pathogens, and mutualistic or beneficial interactions. However, it seems that many researchers assume symbiosis refers to a positive or mutualistic relationship between the host and the symbiont – even though known reproductive parasites such as *Wolbachia* are also frequently referred to as symbionts. We have rewritten to clarify meaning, **lines 32-38.**

L279 - it is hard for me to figure out what is going on from the model, but if the fitness effects of the symbiont were zero, surely it isn't surprising that the fraction with microbes declines if only a subset of this fraction is conveyed by the mother to the next generation? it would be like it having $R_0 < 1$

Response 16: We agree - however we also considered complementation with environmental acquisition and the role of fitness, incomplete maternal transmission could have been offset by increased life history fitness resulting in more 'absolute' carriers. We clarified this in the caption to figure S1 and on lines 259-262; 283-286; 297-299; 304-305; 313-319.

L290 - this set up is rather odd - why not simply have one patch with an introduction rate? or something along the lines of more classic invasibility analysis? or is there a realistic scenario that we are trying to reflect?

Response 17: We set it up this way to try to formalise a question / assumption of many discussions in the literature, which have used observations of mate choice through microbiome changes as evidence of strong precursors of RI. We have clarified the reason for constructing it this way, lines 240-243.

L300 - required for what?

Response 18: Apologies, revised out.

L365 - 'very modest' - relative to what?

Response 19: Thank you, we clarified that rather than host-microbe associations being contingent upon benefits to host fitness and mutualism, our model suggested that, relative to transmission efficiency, there are only very modest changes to host carrier frequency with increasing host fitness, main figure captions, lines 297-299; 373-378.

References

Roughgarden et al. 2018 Holobionts as units of selection and a model of their population dynamics and evolution. *Biological Theory* 13 44-65.

Roughgarden 2018 Model for vertical vs. horizontal microbial colonization bioRxiv:465310.

Vliet & Doebeli 2019 The role of multilevel selection in host microbiome evolution PNAS.

These have now all been added.

Referee: 2

Overall this is a very interesting contribution to symbiosis research. The writing was clear and the background literature was well presented. That mixed mode transmission may be critical for maintenance of symbioses regardless of fitness impacts could be a very important finding, with implications in microbiome, epidemiology, vectored diseases, evolution, and ecology. However, the manuscript is not written for this broader audience and it should be revised to help non-experts understand both how the findings were obtained and their implications, as well as recommendations for how experimental systems could be used to test the predictions of the model.

Response 20: Many thanks for the positive thoughts. We agree that more needed to be done to make the study suitable for a broader audience. We have done this by making the introduction more general and giving a broader context, and doing the same for the discussion, we have extensively rewritten both. In addition, we have better structured the results around the questions and have unpicked the details of the results, plus added summary sentences to give more accessibility, e.g. lines 40-48; figure S1; rewritten discussion and conclusions.

The major criticism I have of the manuscripts pertains to one of the primary, and potentially most important conclusions drawn by the authors: that fitness plays only a "modest" role if any in symbiont transmission. My criticism stems from the fact that the authors seem to downplay the fitness effects that do appear in their models, without sufficient justification or rationale for categorizing this as "modest". For example, the authors chose to separate from each other the four fitness levels tested (side note – the authors should define epsilon and these arbitrary fitness levels for the reader).

Response 21: An important point, the relative effects of fitness were low, hence 'modest' (see also response 22, below), but we appreciate the point that the rationale for downplaying needs to be significantly clarified

and in places the wording changed. We have added sentences to address this point throughout, main figure captions, lines 297-299; 373-378, rewritten discussion and conclusions section.

Fitness levels of 0.75 and 1.33 are shown in one figure, and 1.0 and 2.0 are shown in another. If one examines all four side by side, in increasing order of fitness, a pattern does emerge. Examining only the 0.7 steady state carriage, the intersection of the line with vertical transmission frequency decreases as the fitness increases, and a similar trend occurs on the environmental transmission axis. These trends are consistent with the idea that fitness is important and can counter imperfect transmission. Put another way, the model predicts that a system with higher fitness would require slightly less probability of transmission by either environmental or maternal routes to achieve the same carriage. The difference is slight, but maybe meaningful in an evolutionary context. I think the authors need to do a better job of convincing the reader that this relationship is not significant. It would be interesting to plot the relationship between fitness and steady state carriage reliance on one or the other type of transmission and calculate the slope of that relationship.

Response 22: we agree that fitness effects are present, but over a large parameter space of relative fitness from 0.75 to 2.0, the patterns hardly change. We think this permits a conclusion of modest effects of fitness overall, lines 371-386. However, they could still contribute, as the reviewer suggests, and so we have included a sentence to reflect this in the context of ecological conditions that have yet to be explored, lines 378-381.

On a related point, the wording for the requirement for vertical transmission in inter-population transmission is confusing, and possibly inaccurate. The graphs show that as long as fitness benefits are high (2.0) the perfect vertical transmission is not necessary, but the wording in both the text (L338-340) and the figure legend title for Fig. 2 both indicate that vertical transmission is a pre-requisite to steady state carriage and that benefits only slightly impact it. However, the interpretation is the other way around: benefits can compensate for less-than-perfect transmission. Hopefully that's clear – apologies if not.

Response 23: Apologies for this and we have rewritten it. The organism must be twice as fit as a non carrier for this to occur, but we see this could have been much clearer, figure 1 caption.

A second major concern I have is that the model does not have any verification. What did the authors do to ensure that their formula calculations were robust and not unduly impacted by incorrect assumptions? Are there any existing data that could be plotted to determine if they fit the model? Perhaps aphids, for which there is a wealth of knowledge from both lab and field studies and for which mixed mode transmission has been demonstrated?

Response 24: good point, but as these are questions that we believe have not yet been addressed in their or experimentation we have not been able to find suitable proofing data. So to include this important point, we have added an extra section to explain how we hope that this theory will prove useful as a guide to frame future experiments and that this exercise will also allow the theory to be properly verified, lines 382-386.

On a more granular level, the way the data were presented were difficult to understand and interpret. Of note:

- The supplemental needed more information on how the equations were derived so that an expert could re-evaluate them.

Response 25: agreed, and explanations have been added and the supplement greatly expanded. A link to the Matlab code has also been added for data accessibility.

- Explanation for the degrees of mate choice not described clearly enough in either the text or in the supplemental.

Response 25: agreed, and we rewrote throughout and added extensively to the supplement.

- The mate choice information is not clear. Are the panels in Fig. 1 categorized by male mate choice preference, with the degrees of freedom (1-9) indicating female sampling? I think so from the supplemental but it's not clear enough.

Response 27: apologies, we have greatly expanded this and made it much clearer the form of mate preference, which sex is exerting it, etc.

- Individual panels are dense with information and it becomes difficult to tease apart meaningful from meaningless trends. For example, for the 0.5 steady state carriage line in Figs. 1A and 1B there is only one line apparent (rather than the multiples that are representing all mate choice degrees of freedom). For part of the line (low end of the vertical transmission axis) the lightest shading (equating with 9 of the degrees of mate choice) shows, and presumably all the other lines track with this one. However as soon as the line crosses the vertical transmission frequency of 0.8 it becomes a dark line. Either this is a formatting issue or it has some meaning. Either it has to be fixed or explained.

Response 28: Sorry for the confusion, we have addressed this by playing around with contrasting and alternative colours to increase resolution and clarify meaning. All the figures have been redrawn.

- Supplemental: What are alpha and beta? These may be standard in math models, but for the general audience it will not be clear.

Response 29: Done, see also response 5 and more clearly defined in the caption to figure S1 and in the supplement.

- In Figure 2, when population 1 is the only source of symbionts for transmission to population 2, what is the assumed steady state carriage in population 1?

Response 30: We have now added to the caption the additional explanation that the steady state carriage is illustrated on the left hand panel (e.g. asks what are the parameters required for population 2 to reach the same steady state capacity as population 1).

- Iterative calculation should be explained. I think what they mean is that they have a starting carriage level and at that level determining the assortative mating impact, then back and forth until they reach steady state. But this should not be left up to the reader to try and interpret. How many iterations does it take for the population to reach steady state and did this vary depending on the assumptions/parameters?

Response 31: Apologies, we use the frequencies obtained in one generation to calculate those for the next generation (i.e. each generation depends on the results in the generation preceding it). In the latest version of the supplement we describe it in terms of a two step process - (1) performing all of the mating and inheritance calculations to get proportional frequencies and then (2) normalisation to fill the entire frequency range from 0 to 1 (with social transmission and migration being implemented during phase 2 for ease and timing in the life cycle).

And one last minor comment: In Figure 2 the $n=3$ and 7 are missing from the figure caption, even though they're listed in the legend.

Response 32: Apologies, now corrected.

Proceedings B - Decision on Manuscript ID RSPB-2020-0820

Dear Professor Heesterbeek,

Many thanks for the opportunity to revise our MS, we very much appreciate the opportunity and the supportive and helpful comments from the editor and reviewers. We provide below and point by point description of the extensive changes we have made, as well as a clean and marked up copy of the main MS highlighting the revisions.

We very much hope the MS is now suitable for publication. We can confirm again we have no conflicts of interest and that all authors are aware of, and have approved, this revised submission.

Best wishes
Tracey Chapman
(on behalf of all authors)

Associate Editor comments:

Thank you for submitting your work for consideration in the special issue. Your manuscript has now been read by myself and two reviewers, and while we all see value in the work and a great need for these types of analyses, both reviewers found parts of the manuscript (especially the presentation and visualization of the results) difficult to understand. Given that we all feel this could make a strong contribution to the literature, I would like to ask the authors to revise the work in light of these thoughtful and thorough comments. I look forward to reading the revised manuscript, and thank you in advance for your work.

#1. We thank the editor and reviewers for all the helpful and constructive comments, which we found extremely useful. We appreciate the thought, time and care that went into the reviews. We provide below a point by point description of all revisions made and a marked up MS copy with changes highlighted. We made significant changes throughout, but particularly to the presentation, description and visualization of the results. We added a new table of definitions and to capture how these features were encoded within the model. We hope that this will provide much greater clarity on the major concepts and themes. We have endeavoured to be as thorough as we could, as instructed and believe that with this revision we have captured all the suggested and required revisions.

Referee: 3

The manuscript 'Transmission efficiency drives host-microbe associations' explores and discusses different factors that may play a role in the establishment of host-microbial associations, particularly focusing on modes of transmission. The authors first review the literature on vertical, horizontal and mixed modes of transmission, and then use a theoretical model to assess how combinations of VT and HT lead to host-microbe associations in a population of hosts. This is a very interesting topic and I agree with the authors that a simple theoretical model as used here has the potential to provide great insights in the evolution of host-microbe associations. In its current form, however, I feel that the manuscript largely fails to do so. See below my comments, ordered by section.

Introduction/review

I appreciated the introduction and review of transmission modes, and the authors did a nice job of summarizing a complicated field. It would be helpful to mention the review in the abstract.

#2. We have now mentioned our review of the existing literature in the abstract (from line 4).

There are several places where being more specific about what their terms mean would be helpful. For example, in this paper, 'stability' is the measure that marks an evolutionary host-microbe association. However, the authors never quite define what exactly is meant by stability—is this on the level of a host population? Is it a single microbe? Or some facet/functionality of the whole microbial community? Does it imply some fitness benefit? Especially for the MMT, it seems important to be clear about whether MMT applies to just one microbe, or to the whole community:

Is it like in *Drosophila*, where different microbial species have different transmission modes (VT: *Wolbachia*, HT: *Acetobacter* and *Lactobacillus*), or, is it something like the *Acetobacter thailandicus* example (line 183-186), where the same microbe can have both pseudo-VT and HT modes? Or can it include both? Defining 'stability' more clearly early in the manuscript would help establish the eco-evolutionary scale at which the insights from the model apply.

#3. In line with this and other reviewer comments below – we have revised extensively to try to clarify the focus and also the limitations of our modelling approach. We now define the conditions under which higher population-level frequencies of host-microbe associations occur. Previously we referred to this as 'stability' – a measure of a host-microbe association with significance for evolution. We have now revised this for clarity and greater accuracy by replacing references to 'stable host-microbe associations' with 'high frequency host-microbe associations' throughout the manuscript. Parameters tested as 'stabilising' host-microbe associations are instead 'promoting' the population level frequency of host-microbe associations'. We hope that these revisions made throughout, plus the addition of the new table (see #4 below), serve to better define terms and concepts.

Methods/results

The paper does a poor job in explaining, presenting and visualizing the methods and results. Without reading the SI, it is essentially impossible to understand and interpret the results (and even after reading the SI, I'm still a bit puzzled). For instance, it is unclear in the main text how VT, HT and social transmission are defined in the model, while these concepts are central to all results. Not all details need to be given in the main text, but it would help to give the readers some idea about all variables and processes.

#4. We include a new Table of definitions (Table 1 Line 56) to better describe terms and how they are defined in the model. The new table gives the biological definitions of various transmission parameters that we use, as well as summarising the method via which each term was included within our models.

I am convinced that there must be a better way to visualize the results. I find the figures hard to digest, and not very appealing. At first glance, they basically all look the same, so it is very difficult to see which differences the authors want me focus on (e.g. is there an important difference that I should notice between Fig. 1a and 1b?).

#5. Thank you for this important comment – we agree that figures contain a lot of information and that it has been a challenge for us to get this across simply and clearly. The mechanism for best displaying the results is something that we have explored in depth by trying out different formats, colours and schemes. We have come to the overall conclusion that we have not yet been able to find an alternative format that would work as well as the current in terms of being able to simultaneously display the response to the different parameters varied. That said, we recognise the problem and have made a number of adjustments and changes to the main Figures 1 and 2 to make them easier to understand and specifically to point up the specific features that we discuss in the text. In particular, we have adjusted the figure legend to more clearly define the meaning of different "n" values, ranging from n=1 (random mating) to n=9 (the choosiest case considered). We also added arrows to each of the right-hand panels that point the reader to the main differences within the figures. Figures S2-S6 are also now rendered with larger font to aid readability. A comment reflecting this change has also been added to each figure caption. We also revised part of the colour scheme as we noted there could be difficulties with visualising the lighter colours on some types of display screens. (Lines 314-316 & 367-369). See also #6 and #7, below.

I encourage the authors to think about a better way to visualize their results, better guiding the reader to detect the important patterns. I suggest to add lines for which host variation is zero, i.e. those combinations for which carrier frequencies are either 0 or 1, as these seem particularly interesting.

#6. In the type of mathematical models considered here the microbe carrier frequency will asymptote toward zero rather than ever actually reaching it (unless the microbe transmission/acquisition parameters are all zero). Thus, a carrier frequency line for an equilibrium of zero will not exist. For an equilibrium carrier frequency of one, the line directly overlies the box around the edge of the plotted parameter space (i.e. it requires perfect transmission/acquisition)

and so we conclude that it is not of any practical value for interpreting the results. While these lines have not been added to the figures in the manuscript, we have added a comment to the caption of Figure 1 to clarify this point (Lines 328-331).

Maybe it helps to present the results as a heatmap? Instead of showing two arbitrary values for relative fitness, it would be useful to show how results change as a function of fitness. For instance, a graph showing the relation between relative fitness (x-axis) and the required VT/HT (y-axis) to obtain a carrier frequency of e.g. 0.1, 0.5, 1. I believe that there are more possibilities to 'summarize' some of the results that are now presented in many almost identical graphs (both in the main text and SI), and that such graphs will greatly help to see emerging general patterns.

#7. The figures presented in our manuscript are effectively already simplified heatmaps (they emphasise certain equilibrium carrier frequencies to make comparison between panels simpler). A heatmap 'wash' over the specific carrier frequencies does not make them easier to see. We can't see that adding an extra dimension to these plots by considering variation in another parameter is possible. But as above, we appreciate the importance of the criticism and realise how crucial it is for readers to be able to better observe our results. Therefore, we have altered the figures to emphasise the key points we would like the reader to be able to draw from them. As described in #5, we have now included arrows in the right-hand panels of each figure to demonstrate how the equilibrium carrier frequency lines are altered as a result of increasing the fitness of microbe carrying individuals. We have also adjusted the figure legend to more clearly show the degrees of mate choice considered and to which values of n these correspond.

Perhaps it also helps to start with presenting the results of the 'core model', only considering the effects of VT and HT, and then stepwise add more complexity (reproductive mode, social transmission, dispersal etc.).

#8. This is an interesting point and we see the value of the approach, though the models as currently presented add in and explore the effects of the different factors mentioned in turn in any case (e.g. progression through the figures in the SI). We address this comment by describing the stepwise approach in the text, line 238-257.

Figure S1 is useful, I suggest adding such a figure that explains the modeling procedure to the main text. Please add labels i)-iii) to figure S1, now it is unclear what the caption refers to.

#9. Thanks for this. Labels (i) - (iv) have been added to Figure S1 to more clearly link between the image and the explanation in the caption. Due to limitations of space, we were unfortunately not able to include Figure S1 within the main manuscript. We have however added Table 1 to the main MS, which briefly explains the modes of microbe transmission/acquisition covered in this study and gives a summary of how these are represented within the mathematical model.

Another important point is the way HT (environmental acquisition) is incorporated in the model. The authors write in the SI that HT is defined as '...the probability that an individual will have acquired the microbe in the time period between their birth and their becoming sexually mature.' Both VT and HT thus give the probability that an individual becomes a carrier, either from its mother, or from the environment. Does this imply that VT and HT are essentially the same in the model (the only difference being that non-carriers cannot transmit it to their offspring, irrespective of the population-level VT)?

#10. Thank you for this comment, which prompted a lot of useful thinking on our part. The reviewer is correct that all transmission modes impact the individual's probability of being a carrier/non-carrier before they sexually mature. However, we think there may be some confluences of terms. For example HT is not the same as environmental acquisition in our models and Environmental Acquisition (EA) is often described as *containing* horizontal transmission as well as acquisition from the habitat and diet for biological definitions. We chose to evaluate these factors independently so that we could have a steady-state level of microbial acquisition (EA) and one that is dependent on the population-level carrier frequency (HT). Alternatively, VT is the probability of becoming a carrier if an individual's mother is a carrier and is density independent. Transmission efficiencies for each term can be modelled independently.

Following this prompt to our thinking, we recognised that this indicated we needed to improve the clarity of our definitions, and so this was part of the motivation to add Table 1 (Line 56) to the main MS to more clearly define each of the modes of transmission/acquisition considered within this study, alongside a summary of how each is implemented within the mathematical models. We believe that this will allow the reader to more easily understand the differences between each mechanism and link the different part of the manuscript together more effectively.

In the introduction (L206), the authors write that ‘...MMT will reduce the strength and consistency of VT...’. I’m wondering if the presented model captures this, as VT and HT are not mutually exclusive.

#11. Changed to ... While MMT has the potential to reduce the strength and consistency of VT and selection for tight co-associations between microbes and hosts ... Lines 225-226.

If I understand the modeling procedure correctly, once a microbe is vertically transmitted, a host will never lose it, for instance through horizontal acquisition of competing microbes. Instead, the two modes of transmission act in an additive way, providing two independent routes by which hosts can acquire (but never lose) microbes. It is thus not surprising that increasing either of them, or both of them, all increase carrier frequencies. I am not convinced, though, that the correct conclusion is that mixed modes of transmission lead to more stable host-microbe associations (but rather: more faithful transmission, from VT and/or from HT, leads to more stable host-microbe associations)?

#12. Thank you for this interesting thought. L380 in discussion now reads “Overall the modelling results revealed that the spread of microbes at a high frequency within a host population was more easily attained when there were high fidelity transmission routes, and mixed modes of transmission that incorporated both maternal VT, HT, and environmental acquisition.” Which we think reflects better the approach that any high-fidelity transmission would promote host-microbe associations, but that MMT has been underrepresented.

I think this also relates to the somewhat unexpected result that relative fitness has a relatively small impact on the frequency of carriers. The probability of getting the microbe (via either HT or VT) is fixed, so new carriers are being re-introduced every generation as determined by their HT/VT, no matter their fitness consequences. In natural populations, however, I would expect there to be selection acting on HT/VT directly (e.g. where hosts would evolve ways to avoid the exposure to certain microbes).

#13. Again, thank you. We now explicitly propose experiments to “... test the assumptions in our model that different transmission modes are routinely additive, when multiple species/strains of microbes are considered, do these instead become routes for microbe-microbe exclusion and competition?” L449

Although I think it is perfectly fine that this paper focuses on equilibrium frequencies given fixed transmission patterns, instead of on the evolution of modes of transmission,

#14. We now refer to fixed frequencies of association throughout the text – in recognition of this point.

I think this should be made clearer throughout the text, through more careful formulation (for instance on L392: ‘We...found that increases to host fitness...had only minimal effect on promoting host-microbe relationships’. Is this a valid conclusion? Host-microbe relationships are not promoted, but selection simply cannot get rid of them). Defining different terms more precise (see comments above), will help.

#15. We agree that this previous text was slightly ambiguous and we removed it from the manuscript, as the statement in the previous paragraph better summarised our findings. Line 406 onwards “Surprisingly, our results also suggested that the relative fitness of host carriers vs non-carriers was less important for increasing host microbe carriers than the existence of efficient microbial transmission [1]. This is not to say that effects of microbes on the fitness of their hosts were absent. However, over a large parameter space of relative fitness from 0.75 to 2.0, the frequency of host-microbe carriers hardly changed” We have also made sure that fitness is defined in biological and modelling terms in Table 1.

Specific comments about the current figures:

1) Both in figure 1 and figure 2: please explain what additional degrees of host mate choice means i.e. is $n=1$ the choosiest and $n=9$ less choosy?

#16. Yes, thanks for spotting this. Figure legends have been altered to give the interpretation on the degrees of mate choice considered – from $n=1$ (random mating) to $n=9$ (the choosiest). The caption of each figure also now states that “Multiples of each line represent additional degrees of male mate choice, e.g. the number of opportunities a male has to find a preferred mating partner from $n=1$, (no choice) to 9 (choosiest).”

2) Figure 2: population 1 and 2 should be labelled on the figures.

#17. Thank you, these labels have been added to figures in both the main text and supplementary information.

3) No need to add a legend to each panel, but do add a title to the legend.

#18. We weren't quite sure what this referred to, specifically, but have neatened up the figures by switching to a single legend for each figure (rather than one for each panel). This has also been expanded to make clear the meaning of the degrees of mate choice considered – from $n=1$ (random mating) to $n=9$ (the choosiest).

Discussion

The discussion could benefit from additional contextualization of the results. In its current form, the review and modeling part of the manuscript are largely disconnected, and I was hoping that the discussion would bring the two together. For example, the results from the migration model are quite interesting—where assortative mating can modulate the reinforcement or relaxation of reproductive isolation across populations (L320-328). The authors repeat these findings in the discussion (L401-403), but it would be interesting to compare these results to some of the examples discussed/cited in the review section.

#19. We agree, and from **Line 435** it now reads ‘This suggests that direct effects of microbes on host mate choice cannot themselves result in reproductive isolation in otherwise homogeneous populations but could reinforce or breakdown pre-existing genetic isolation, depending on whether non-carriers prefer to avoid or seek out carriers (i.e. the form of host mating preference), and there is strong fidelity of microbial transmission. This is an appreciation of microbial transmission dynamics as a keystone to the model of microbe-induced RI not considered in previous reports [46], but see [31], which is vital.’

While I understand this would mostly be speculation, I think a little bit of connection would help the reader understand how to apply their theoretical results to natural systems. The authors suggest that future experiments could be used to test the modeling predictions (L382-386), but it remains unclear what kind of data/experiment one would need.

#20. Yes, we agree this is useful. We explicitly suggest several experimental approaches now on **Lines 441-452** as follows:

“We propose testing whether modest levels of VT, within these partially isolated populations, where key bacteria are be routinely ‘added back’ to their respective original populations, could establish long-term associations where the introduction of competing microbes is restricted by host-mating exclusion. However, from our models, this effect appears to be moderated by the relative strengths of HT and environmental acquisition for microbe associations with juvenile hosts, and this could be tested empirically by combination approaches of microbiome community analyses and/or labelled bacterial strains to test their relative strengths in different model systems. Similar approaches could also be able to test the assumptions in our model that different transmission modes are routinely additive, when multiple species/strains of microbes are considered, do these instead become routes for microbe-microbe exclusion and competition?”

The discussion of fitness effects warrants some additional contextualization. For example, L371-373, the authors connect phenotypic and behavioral effects to ecological conditions that promote a

high frequency of carriers (is this different from stability? And is this really 'promoting', see my comment above? Another place to where fuzziness impedes the reader). The model doesn't really incorporate differences in ecological conditions, unless this is to be implied by the different populations? Perhaps application to empirical data would be helpful to better explain what is meant here. As stated earlier, I think this study could be a valuable contribution to our understanding in host-microbe associations, but a little bit more context is needed in the discussion to apply the theoretical findings.

#21. Agreed, **Line 400** now reads "The results suggested that neither phenotypic nor behavioural changes in the host (e.g. due to host mate choice for carriers) had a significant bearing on the transmission efficiencies required to promote a high frequency of host microbe carriers".

We also include on **Line 414** "If there is a strong fitness benefit to an association, it is possible that there could be selection on higher fidelity of transmission, and future research should also seek to verify these models, introduced fluorescently labelled bacteria with whole community analyses could be an easy mechanism to track transmission and fitness benefits across generations, to check that the reported outcomes are not unduly impacted by inaccurate model assumptions"

Referee: 4

This manuscript studies the evolutionary stability of host-microbe associations focusing on the efficacy of three types of microbe transmission modes: environmental acquisition, vertical transmission and horizontal transmission. The authors found that mixed modes of transmission (MMT), mostly involving environmental acquisition and vertical transmission are more efficient in stabilising host-microbe associations across generations than each transmission mode alone. The authors review the literature of microbe transmission modes and discuss the relevance of their findings.

In my opinion, the manuscript provides an important contribution to the field, as it improves our understanding of the interplay between microbe transmission modes, with implications to reproductive isolation and the persistence of associations after dispersal. However, in my opinion, the manuscript still needs to be improved in some respects. Below I describe my major and minor concerns in detail:

Major concerns

The meaning of «stabilizing», «stable» and «persistent»

The authors use these words throughout the manuscript, but they are not explicit as to their meaning, which can be twofold: it may be the transition from transient to permanent infection at the individual level; but it can also be the persistence of infection over evolutionary time, that is, the transition from transient to permanent infection at the population level and across generations. It may not be immediately obvious to the reader that these concepts refer to the persistence of infection at the population level. So, I advise the authors to make this clearer in the text, especially in the abstract and introduction, where the concepts appear for the first time.

#22. Agreed, this was also raised by reviewer 3, and we believe we have now addressed this in #3, #4 & #10 above.

The concepts of Environmental Acquisition (EA), Vertical Transmission (VT), Horizontal Transmission (HT) and Social Transmission (ST)

The manuscript is not coherent about the use of these concepts. First, ST appears only a few times in the main text but appears frequently in the supplementary material, including in all figures. Second, ST is combined in the main text with the other transmission modes, while in the supplementary material it is shown as an independent mode of transmission. These discrepancies make the manuscript confusing. If ST was tested as an independent transmission mode, it should appear in the main text as such. Besides, ST, VT, HT and EA are not synonyms. EA should not be called HT because it is not a transmission mode between social agents but between animals and their external (asocial) environment; ST, on the other hand, can be either vertical, horizontal or oblique if it is between parents and offspring (VT), older and younger individuals not directly related (OT) or between individuals of the same age class (HT). Because the models that were here developed simulate discrete generations, there is no OT and so this term can be omitted. However, for the remaining concepts, I advise the authors to use them in a more systematized

way: use ST only as a general term for VT and HT and use VT, EA and HT in all other cases, including the model parameter and all the figures.

#23. This is a very important point, also picked up by reviewer 3. We extensively reviewed our use of terminology throughout the manuscript, and made sure to use it consistently and in line with the reviewer's suggested parameters. For clarity we have included a list of terms and definitions in Table 1, see also #3, #4, above.

This also implies that the authors change the second section of the introduction (lines 91-165): this section should be only about EA, and the authors should include a new section only about HT. Finally, the section about MMT (lines 167-212) should include references to all three transmission modes: VT, EA and HT.

#24. We have made the changes as suggested on lines (Lines 101-184), EA and HT are still in a section together as many of the examples in the literature cannot be reliably allocated to one mechanism or the other, but we have clearly demarcated the two terms in separate paragraphs. We have also made sure to reference all three transmission modes when discussing MMT (Line 208).

Evolutionary significance of host-microbe associations

I do not fully understand the evolutionary component of the mathematical models the authors developed. Although the models simulate steady-state microbe carrier frequencies, I suppose the populations reached those frequencies after several generations. But this is not visible in the figures and is also difficult to extract from the description of the models in the Supplementary Material (SM).

#25. We have been through the paper and revised our emphasis on evolutionary significance – and instead focused on sharpening our presentation of what the models really demonstrate, which is population level microbe-carrier frequencies. We do find a large variation in the number of generations taken to reach a high carrier frequency which depends on parameter combination and also the type of mating preference considered. Comments detailing this have now been added at lines (Lines 301-306) to describe the time taken to reach a microbe carrier frequency of >0.9 .

To improve the manuscript on this respect, the authors could provide the simulation algorithm, the sequence of steps since the creation of the simulated world, with a certain number of individuals of each type, until reproduction, migration, survival, etc. What kind of world it is: a grid with patches or a unique patch? What is the sequence of steps? How many individuals begin the simulations and how many survive each generation? How many generations? Unless there is a method section (and not just a supplementary method section) that I did not have access to, this information needs to be added to the manuscript.

#26. We have altered the sentence at Lines 236-270 of the main text to more clearly describe the type of mathematical model used in this study. We have also made similar changes to the opening section of the Supplementary Text to explain the type of mathematical models used earlier in the document. In particular, we have clarified that this model considers the frequencies of carrier and non-carrier individuals in a deterministic mathematical model of a well-mixed population. We believe that this makes the understanding of subsequent sections outlining the exact model equations considered more instantly accessible.

Minor concerns

Lines 35-36: Here, the authors could mention that microbes can also BE SOCIALLY TRANSMITTED.

#27. Done

Line 57: At the end of that sentence, the authors could state they will also review «the as yet under-researched evolutionary potential for mixed modes of transmission».

#28. Done

Line 59: «Heritable» is redundant with «vertical transmission». Besides, microbes transmitted horizontally are also heritable.

#29. Done

Lines 88-89: Please, specify that coprophagy and social acquisition are between mother and offspring in that case. This is important because horizontal transmission is also a type of social transmission.

#30. Done

Line 115-116: From my understanding, this is not a case of EA but HT. This example should be in a separate section dedicated to HT.

#31. Done

Lines 100-107: These two sentences provide contradictory evidence about the co-occurrences between microbial communities and hosts. Although these are two possible scenarios, they are not logically interconnected and seem to contradict each other. I suggest pulling these two sentences in the same paragraph and interconnect the two ideas better.

#32. Done

Lines 107-117: Shouldn't this paragraph be at the beginning of section (ii)?

#33. This section has been restructured in line with reviewer comments and is introduced earlier in the section.

Lines 124-125: That sentence should have a reference, and it would be clearer if the authors explain how transient gut microbiota can cause RI.

#34. Done, as follows:

"It has been proposed that influences on host mate choice effects by transient gut microbiota, such as a preference for carriers of a particular microbe to only mate with other carriers of the same microbe, can act as a precursor to reproductive isolation and thus speciation." And we have added the reference (Bacteria-induced sexual isolation in *Drosophila* - <https://doi.org/10.4161/fly.5.4.15835>)" **Lines 138-141**

Line 168: The authors could write «highly specialised INTRACELLULAR host-microbe associations». This will help to contrast with the gut microbiome system, that is not intracellular, in the next paragraph (line 179).

#35. Done

Lines 189-191: I do not understand that sentence. How can parent-offspring transmission occur if it is mediated by independent replication of microbes in the environment?

#36. We have changed this at **Line 96-99**

"Despite this, a number of direct and indirect routes for VT have been identified, including (i) contact smearing of microbes onto the egg surface during or after oviposition [13], (ii) oviposition site inoculation and reingestion by offspring [14], (iii) coprophagy [15] and (iv) social acquisition from parent to offspring [16]."

Line 220: «established residencies with hosts, which then spread at a population level» AND ACROSS GENERATIONS, right?

#37. Thank you, done

Lines 226-227: This is just a detail, but it would be better if the authors recapitulate the modes of transmission they considered for their models (VT, EA, HT and MMT) and then inform that they were tested either singly (VT, EA and HT) or in combination (MMT).

#38. Done **Line 245**

Lines 228-230: The authors simulated two populations to test reproductive isolation, but it is missing from the main questions/goals. They could add that goal to the second question.

#39. Thanks for spotting this. We have included point (iv) in our main goals (iv) do parameters alter when considering homogenous population spaces when compared to partially isolated populations? **Line 52.**

Lines 397-411: The results the authors obtained with social transmission (which they should call HT) were not discussed. I think a discussion of these results should be included here.

#40. This has been corrected throughout and is also addressed at Lines 354-358

Titles of figures 1 and 2: The titles of the figures are very different from each other. The title of figure 2 is a result and that of figure 1 is a description of the model parameters. It would make more sense for both figures to have similar titles. My advice is that the titles are based on the description of the model parameters. And, in turn, the subtitles of section 2 (lines 253-255 and 307-309) are a description of the results (like in figure 2).

#41. Done – we have replaced figure 1 legend title with

'High frequency transmission of microbes (through a variety of mechanisms - MMT) is a primary determinant of host-microbe carrier frequencies within a single population of sexually reproducing individuals.'

Legends of figures 1 and 2: The legend of the figures should include a brief explanation of the meaning of n in «the degrees of host mate choice (n= 1,3,5,6 or 9)».

#42. The legends for Figures 1 and 2 have been adjusted to provide more clarity. In particular, we have added the title “mate choice” and added labels “random mating” for n=1 and “choosiest” to n=9.

Figures from the SM: The letters of the graph axes are too small.

#43. Figures S2 – S6 now use a larger font size throughout to make axis labels more readable.